**Assessing Typhoon Soulik-induced morphodynamics over the Mokpo coast region in South Korea based on a geospatial approach**

Sang-Guk Yum[1], Moon-Soo Song[2], Manik Das Adhikari[3*]

[1]  Department of Civil Engineering, Gangneung-Wonju National University, Gangneung, Gangwon-do 25457, South Korea; skyeom0401@gwnu.ac.kr
[2]  Department of Safety & Disaster Prevention Engineering, Kyungwoon University, Gumi, Gyeongsangbuk-do 39160, South Korea; songms0722@ikw.ac.kr
[3]  Department of Civil Engineering, Gangneung-Wonju National University, Gangneung, Gangwon-do 25457, South Korea; rsgis.manik@gmail.com

*Correspondence to: Manik Das Adhikari (rsgis.manik@gmail.com)*

**Abstract**

The inner shelf and coastal region of the Yellow Sea along the Korean peninsula are frequently impacted by Typhoons. The Mokpo coastal region in South Korea was significantly affected by typhoon Soulik in 2018, the deadliest typhoon strike to the southwestern coast since Maemi in 2003. Typhoon Soulik overran the region, causing extensive damage to the coast, shoreline, vegetation, and coastal geomorphology. Therefore, it is important to investigate its impact on the coastal ecology, landform, erosion/accretion, suspended sediment concentration (SSC) and associated coastal changes along the Mokpo region.

In this study, net shoreline movement (NSM), normalized difference vegetation index (NDVI), fractional vegetation coverage (FVC), coastal landform change model, normalized difference suspended sediment index (NDSSI), and SSC-reflectance relation have been used to analyze the coastal morphodynamics over the typhoon periods. We used pre-and post-typhoon Sentinel-2B MSI images for mapping and monitoring the typhoon effect and recovery status of the Mokpo coast through short and medium-term coastal change analysis. The findings highlighted the significant impacts of typhoons on coastal dynamics, wetland vegetation and sediment resuspension along the Mokpo coast. It has been observed that typhoon-induced SSC influences shoreline and coastal morphology. The outcome of this research may provide databases to manage coastal environments and a long-term plan to restore valuable coastal habitats. In addition, the findings may be useful for post-typhoon emergency response, coastal planners, and administrators involved in the long-term development of human life.

**Keywords:** Typhoon Soulik, Coastal changes, NDVI, FVC, Suspended sediment movement,

Shoreline change, Mokpo Coast.

## 1. Introduction

Typhoons are one of the most destructive natural calamities. Strong winds that accompany typhoons damage the environment, coastline, wildlife, people, and public and private properties in coastal and inland areas during landfall (Shamsuzzoha et al., 2021; Xu et al., 2021; Mishra et al., 2021a; Nandi et al., 2020; Sadik et al., 2020; Sahoo and Bhaskaran, 2018; Hoque et al., 2016). Many coastal and near-coastal countries are plagued by typhoon-induced storms, flooding, deforestation, and increased soil salinity (Rodgers et al., 2009). Typhoons (tropical cyclones) have caused 1,942 disasters in the past 50 years, resulting in 779,324 fatalities and USD 1,407.6 billion in economic losses worldwide (WMO, 2020), demonstrating their effects on both the global and regional economies (Bhuiyan and Dutta, 2012; Mallick et al., 2017). The effects of typhoons include saltwater intrusion, soil fertility depletion, reduced agricultural productivity, life losses, coastline erosion, vegetation damage, and massive economic disasters (Mishra et al., 2021b).

According to instrumental data collected since 1904, typhoon intensity on the Korean peninsula has grown during the previous 100 years (Yu et al., 2018; Cha et al., 2021). A total of 188 typhoons, about three annually, have affected the coastal region from 1959 to 2018 (KMA, 2018). Among past Typhoons, RUSA (2002), MAEMI (2003), NARI (2007), and SOULIK (2018) heavily affected the southwestern coast, causing extensive damage to lives and properties (KMA, 2011; 2018). Furthermore, people living in these regions have faced serious coastal floods caused by these events for more than a half-century (Moon et al., 2003). Mokpo coastal region, located in the southwest coast of South Korea, has been hit by 58 typhoons since 1980, with most occurring in the July to October period (Kang et al., 2020; Lee et al., 2022). The rapid growth of coastal economies and populations in recent years has made these areas more susceptible to typhoon disasters. Therefore, the increasing frequency of typhoons on the southwestern coasts is a significant issue for disaster management.

Several studies (Halder and Bandyopadhyay, 2022; Wang et al., 2021; Shamsuzzoha et al., 2021; Kumar et al., 2021; Sadik et al., 2020; Konda et al., 2018; Parida et al., 2018; Zhang et al., 2013; Yin et al., 2013; Li and Li., 2013; Rodgers et al., 2009) have been carried out in South Asia using various techniques to map the hazard, vulnerability, risk and effects of typhoon disasters. Remote sensing and geospatial technology play a crucial role in monitoring a variety of natural disasters (Wang and Xu, 2018; Mishra et al., 2021b; Charrua et al., 2021).

The majority of studies on typhoon-induced coastal dynamics rely on passive optical remote
sensing and identify natural disaster damage using changes in landuse data, vegetation indices,
and geospatial techniques (Mishra et al., 2021a; Xu et al., 2021; Nandi et al., 2020). The post-
typhoon damage assessment research in South Korea mostly focused on property loss,
economic losses, and casualties (Yum et al., 2021; Kim et al., 2021; Hwang et al., 2020).
However, the coastal morphodynamics along the Mokpo coast over the typhoon period (such
as short and medium term) have not been investigated in detail. Thus, this study's primary focus
is to determine the effects of typhoon Soulik on coastal ecology, landforms, erosion/accretion,
suspended sediment movement and associated coastal changes along the Mokpo coast.
The normalized difference vegetation index (NDVI) and variations in NDVI (ΔNDVI)
have been used to map the extent of vegetation destruction and details on the degree of damage
after the typhoon (Wang et al., 2010; Datta & Deb, 2012; Zhang et al., 2013; Kumar et al.,
2021; Xu et al., 2021). Vegetation damage can be seen by the negative change in NDVI values
between the post-and pre-typhoon period (Mishra et al., 2021a; Hu and Smith, 2018). On the
other hand, fractional vegetation coverage (FVC) is a crucial quantitative indicator of the
vegetation cover of the land surface (Zhang et al., 2021; Wang and Xu, 2018; Song et al.,
2017). Therefore, FVC has also been used to assess the extent of vegetation damage caused by
typhoon Soulik and to analyze its impact on vegetation cover. The coastline movement over
the typhoon periods has been analyzed using the Digital Shoreline Analysis System (DSAS)
program (Tsai, 2022; Adhikari et al., 2021; Bishop-Taylor et al., 2021; Santos et al., 2021). In
order to monitor and protect coastal habitats, we need to understand the distribution and
movement of SSC between rivers and coastal waters. Thus, the normalized difference
suspended sediment index (NDSSI) (Kavan et al., 2022; Shahzad et al., 2018; Hossain et al.,
2010) and the SSC-reflectance algorithm developed by Choi et al. (2014) for the Mokpo coastal
region have been used to monitor SSC distribution. Furthermore, to understand the short and
medium-term morphodynamics of the coastal landform due to the typhoon, a GIS-based coastal
change model has been developed. Four coastal landform classes, i.e., tidally influenced land
(wetland land, wetland vegetation) and non-tidally influenced land (land and water), have been
used for the coastal morphodynamic analysis (Maiti and Bhattacharya, 2011). The change
detection technique has been employed to quantify the short and medium-term coastal changes.
This approach focuses on details of morphological changes within the coast and highlights the
minor changes caused by the typhoon.
This study uses Sentinel-2 MSI images as a primary data source to examine the
morphodynamics and effects of Typhoon Soulik on coastal ecology. Accordingly, the
objectives of this study are to (i) quantify and mapping of coastal landform dynamics prior to
and after the typhoon, (ii) examine shoreline movement and assess coastal erosion and
accretion, (iii) assess the degree of typhoon damage to vegetated land, and (iv) analyze changes
in SSC and the response of sediment dynamics over the typhoon period. Coastal managers can
use this study to develop and implement appropriate strategies and practices to protect natural
ecosystems and post-disaster rehabilitation.
**2. Study Area**
The Mokpo coast is located in the southwestern part of South Korea and is characterized by
muddy flats with wide tidal ranges (Choi et al., 2007; Kang et al., 2007), as depicted in Figure
1. The inner part of the coast includes harbor and industrial complexes, a large residential area,
and a wastewater treatment plant. Mokpo coast is most frequently hit by typhoons, which cause
the most significant amount of property damage and loss of human lives (Kang et al., 2020;
Lee, 2014). According to storm surge records, the Mokpo coastal region has experienced the
highest number of typhoons (58) since 1980 due to its geographical location (Lee et al., 2022;
Kang et al., 2020). The tidal range has been observed to be broader, with the extreme high tide
60cm higher and the extreme low tide 43cm lower in the Mokpo coast (Lee et al., 2022; Kwon
et al., 2018). This fluctuation resulted in significant flooding during the typhoon period. High
water and waves severely damage the coastal structures and environment, especially during
surges (Tsai et al., 2006). The Mokpo coastal region is characterized by a strong ebb dominant
pattern because of its complex bathymetry, scattered islands and extensive tidal flats (Byun et
al., 2004; Kang and Jun, 2003; Kang, 1999).

The vast tidal flat of the Mokpo coast serves as a habitat for many different species, has
a large production capacity, and is highly regarded for its role in cleaning up pollution and
controlling floods and typhoons (Lee et al., 2021; Na, 2004). Furthermore, the powerful storm
has affected the coastal wetlands (mudflats) that serve as the primary spawning and nursery
grounds for fish and other marine life. However, Choi (2014) observed that tidal flat systems
in the Korean peninsula are actively responding to various phenomena, such as tides, waves,
and typhoons. The wetland, coastal vegetation and coastline along the Mokpo coastal region
have been disturbed due to the extreme climatic events. It has been observed that most typhoon
passages severely impacted the tidal flat environment and caused morphodynamics along the
Mokpo coast.

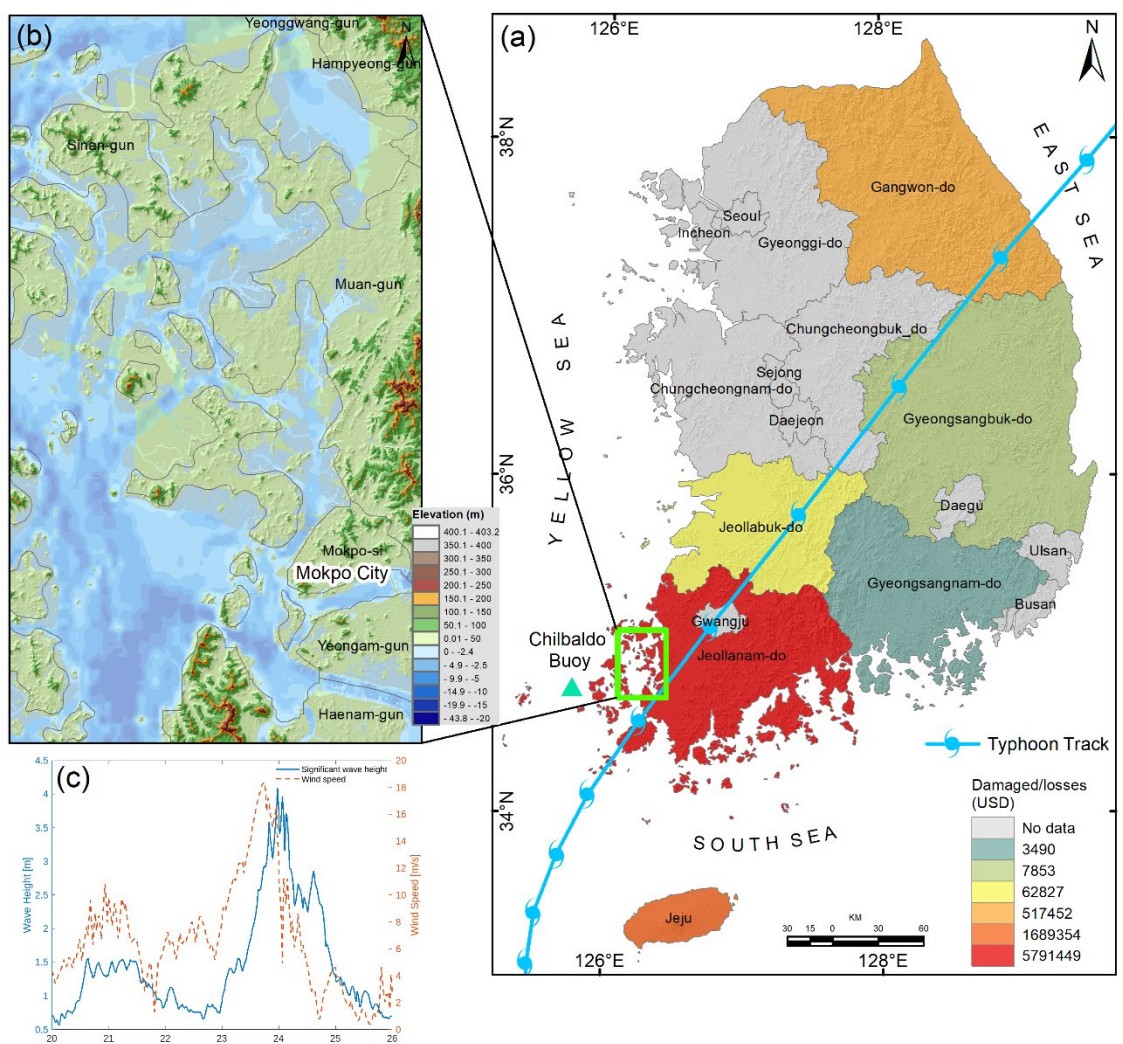


Figure 1. (a) Typhoon Soulik passage passed through the Mokpo coastal region on 23ʳᵈ August 2018 (Typhon track data were downloaded from https://www.ncdc. noaa.gov/ibtracs/), while the background shades represent province-wise recorded damaged/loss distribution reported by Member Report (2018), (b) Topography variation of the Mokpo coastal region (elevation data acquired from NGII (2018), https://www.ngii.go.kr/, and bathymetry data downloaded from GMRT, https://www.gmrt.org), and (c) Variation of significant wave height and wind speed from August 20 to 25, 2018 recorded by Chilbaldo Buoy Station (located near the landfall area) during the typhoon Soulik (Data source: http://wink.kiost.ac.kr/map/map.do# ).

## 2.1 Typhoon Soulik

The southwestern coast of the Korean peninsula was ravaged by the strong intensity typhoon Soulik, which hit the Mokpo coast on 23ʳᵈ August 2018 (Ryang et al., 2021). On 16ᵗʰ August, it developed near Palau as a tropical depression. Subsequently, it strengthened into a tropical storm before intensifying into a typhoon (Lee et al., 2022). It moved into the East China

Sea on 20$^{th}$ August with a maximum intensity of 950 hPa (44 m/s) and lasted until 22$^{nd}$ August.
The Korea Meteorological Administration (KMA) issued typhoon warnings, and national and
local authorities took preventative measures to limit potential damage. On 23$^{rd}$ August, around
14 UTC, Typhoon Soulik made landfall close to Mokpo city, located on Korea's southwest
coast. The typhoon remained on the mainland for an additional 12 hours before moving to the
East Sea, where it underwent a transformation and became an extra-tropical cyclone. A peak
sustained wind speed of 30.2 m/s was recorded at Gageodo in South Jeolla Province, while the
central pressure of the typhoon was measured at 975 hPa (Member Report, 2018). Meanwhile,
the strongest gust was observed at Mt. Halla, with a peak gust of 62 m/s. It also dumped
tremendous rain (Kang and Moon, 2022; Kang et al., 2020; Yu et al., 2018; Cha et al., 2021).
The buoy station near Jeju Island has recorded extreme sea surface conditions, including a
maximum wave height of 15m, gusts of 35 m/s, and a drop in water temperature of 10°C. (Kang
et al., 2020; Yoon et al., 2021). Figure 1(c) illustrates the variations in sea surface parameters
between August 20 and August 25, 2018, in the vicinity of the landfall region (Chilbaldo buoy),
including wind speed and significant wave height. It was observed that a significant wave
height, i.e., 4-6 m, was recorded at Chilbaldo Buoy station. According to the Ministry of the
Interior and Safety (MOIS), typhoon Soulik caused various damages and disruptions across
various regions in the country. One woman was reported missing in the coastal area of Jeju,
and three people sustained injuries. A total of 362 facilities were damaged. In addition, the
typhoon resulted in power outages for 26,830 houses and flooding that affected over 3,063
hectares of farmland (Member Report, 2018). Furthermore, the typhoon destroyed extensive
vegetation with strong gusts and damaged non-residential structures along the Mokpo coast. A
province-wise breakdown of the damage and losses caused by the typhoon is depicted in Figure
1(a). The total damage caused by Typhoon Soulik in South Korea was \$45 million (KMA,

2018).


**3. Data and Methods**
**3.1 Data Sources and pre-processing**
Typhoon-induced coastal dynamics along the Mokpo coast have been studied using the
pre-and post-event Sentinel-2 MSI images. A multispectral instrument (MSI), Sentinel-2,
consists of two polar-orbiting satellites, Sentinel-2A and Sentinel-2B, launched in June 2015
and March 2017, respectively (ESA, 2020). The Sentinel 2 MSI has a 290 km wide field of

view, a minimum revisits period of five days, 13 spectral bands ranging from visible to shortwave infrared (SWIR), and spatial resolution of 10m (4 bands), 20m (6 bands), & 60m (3 bands) (ESA, 2020). The Sentinel-2 User Manual describes the MSI's radiometric, spectral, and spatial characteristics (ESA, 2020).

The cloud-free Sentinel-2 MSI level 1C satellite images with a relatively fine spatial resolution (10m) for the pre-and post-typhoon period have been downloaded from the Copernicus Scientific Data Hub (https://scihub.copernicus.eu/dhus/) as depicted in Figure 2. Level 1C is a 12-bit radiometric product that was presented the top of the atmospheric reflectance value (Phiri et al., 2021). The open-source software SNAP (Sentinel Application Platform) has been used to process the Sentinel-2 MSI images such as masking, band visualization, atmospheric correction etc. We used SANP's iCOR tool (image correction for atmospheric effect) for atmospheric correction of the Sentinel 2 MSI data over the land and water (Tian et al., 2020; Keukelaere et al., 2018). After that, satellite remote sensing reflectance ($R_r$) images were used to monitor short and medium-term coastal dynamics in the Mokpo coastal region.

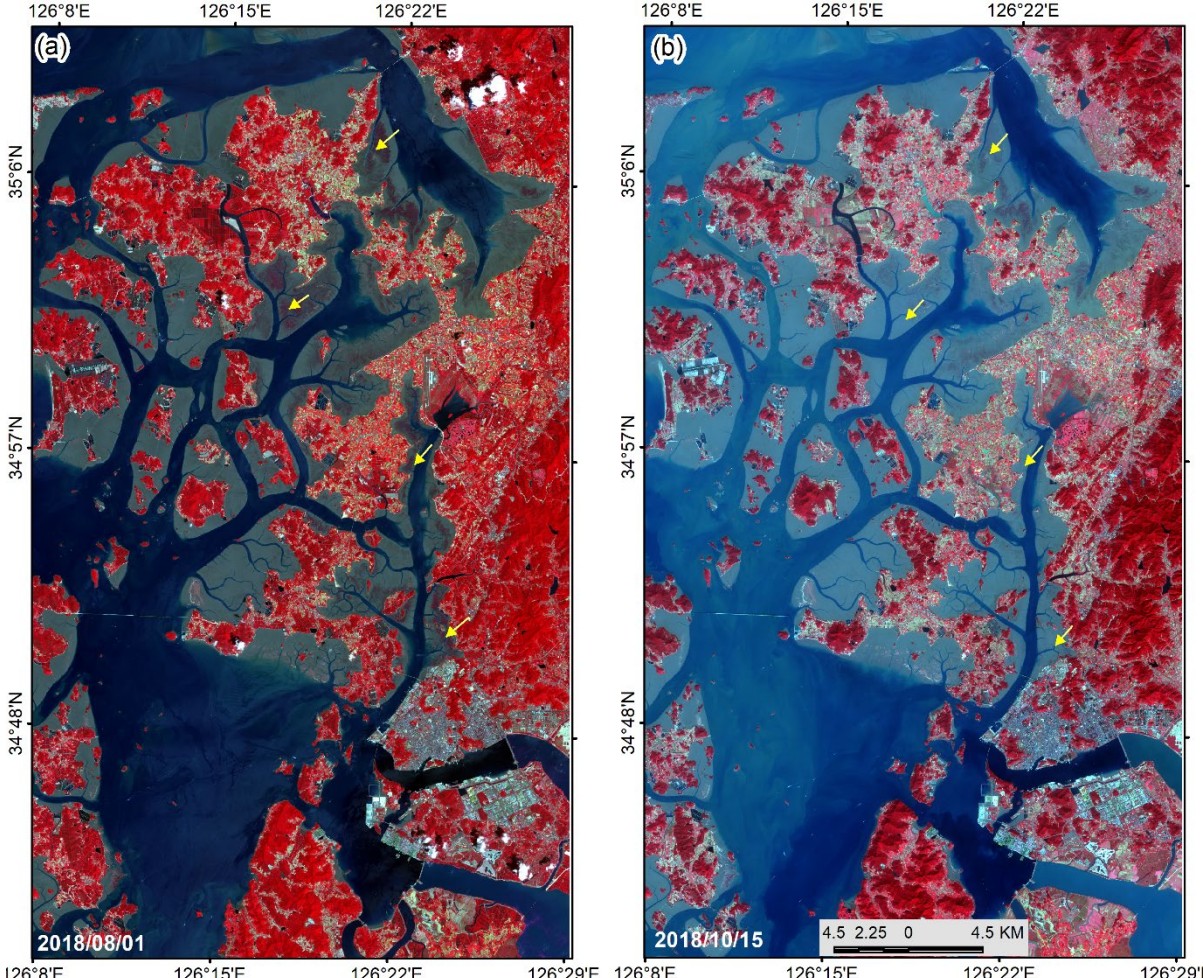

Figure 2: Pre (a) and post-typhoon (b) standard false color composite of reflectance image of the Mokpo coastal region (Sentinel-2 MSI level 1C satellite images are downloaded from https://scihub.copernicus.eu/dhus/). The arrows indicate extensive vegetation damage due to typhoon Soulik.

On the other hand, to exclude the impact of tidal changes, satellite images have been chosen during low tide conditions (Maiti and Bhattacharya, 2009). The tide height has been computed using the WXTide32 program (Hopper, 2004). Several researchers have discussed the significance of low tide satellite data for coastal mapping and dynamics modeling (Nayak, 2002). The details of pre- and post-typhoon satellite data used in the study are listed in Table 1. In addition, the coastal morphology was also investigated using high-resolution (5m×5m) topography data (i.e., LiDAR DEM) provided by the Korean National Geographic Information Institute (NGII) and bathymetry data obtained from GMRT (https://www.gmrt.org) (Fig. 1b).

Table 1. The details of Sentinel-2 MSI data used for coastal dynamic modeling.

| Periods | Date of acquisition | Sensor | Cloud cover (%) | Tidal Height (m) |
|---|---|---|---|---|
| Pre-typhoon | 2018/08/01 | Sentinel-2B MSI | 1.3464 | 0.77 |
| Post-typhoon | 2018/10/15 | Sentinel-2B MSI | 0.6548 | 1.01 |
| | 2019/10/20 | Sentinel-2B MSI | 2.8444 | 1.02 |

## 3.2. Typhoon-induced coastal dynamic modeling

The present study addresses the typhoon Soulik-induced morphodynamics over the Mokpo coast region, specifically examining short and medium-term coastal changes. Short-term coastal erosion refers to the rapid erosion processes and coastal alterations that occur immediately after typhoons or over short durations, typically within days, weeks, or months. Contrarily, medium-term coastal change refers to erosion processes and coastal changes that take place over a period of time ranging from a few months to a few years. It involves the restoration and stabilization of coastal land surfaces after the typhoon. Figure 3 depicts an integrated flowchart of the impact of a typhoon on a coastal system. The outline of the study is divided into four sections: (a) coastal vegetation disturbance mapping, (b) coastal landform mapping and change analysis, (c) suspended sediment concentration variation modeling, and (d) coastal erosion and accretion analysis. The details methodology of each objective has been discussed in the subsequent section.

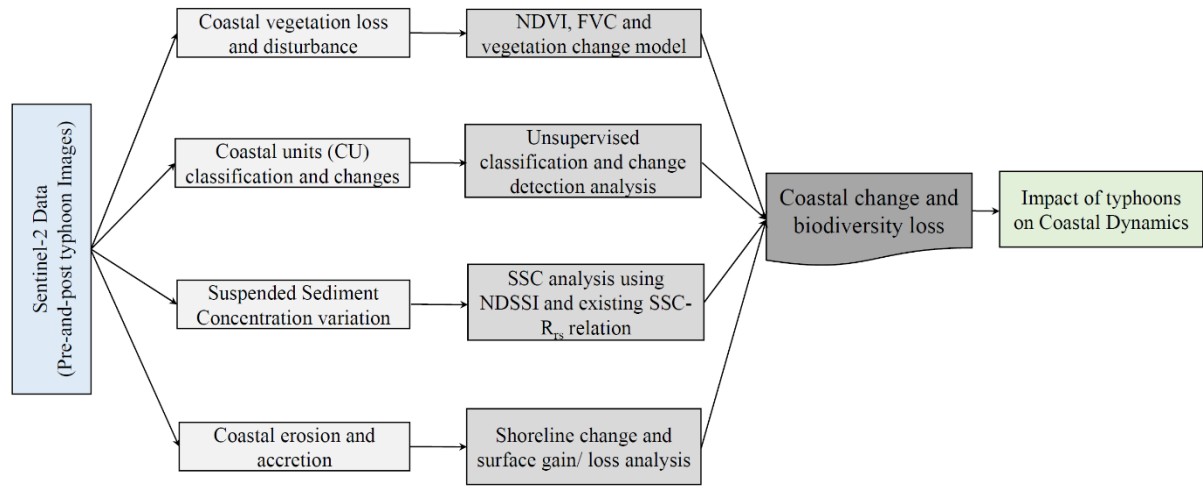


Figure 3. Geospatial-based approach for typhoon-induced coastal dynamics analysis.

### 3.2.1 Analyses of coastal vegetation loss and disturbance

Vegetation damage severity mapping (VDSM) has been performed using pre-and post-
event satellite images. NDVI and FVC are widely used techniques for measuring vegetation
density, health status, regional vegetation condition, and detecting vegetation disturbances (Xu
et al., 2021; Mishra et al., 2021b; Wang et al., 2010; Yang et al., 2018, Wang and Xu, 2018;
Carlson and Ripley, 1997). Subsequently, numerous studies (Xu et al., 2021; Mishra et al.,
2021a; Charrua et al., 2021; Shamsuzzoha et al., 2021; Kumar et al., 2021; Nandi et al., 2020;
Wang and Xu, 2018; Konda et al., 2018; Zhang et al., 2013; Rodgers et al., 2009) have shown
that the NDVI and FVC is a reliable indicator of post-typhoon damage detection. Therefore, in
this study, the vegetation damage due to typhoon Soulik has been determined using the NDVI
and FVC approach. The NDVI has been calculated by using the following Eq. (1) (Rouse et
al., 1974; Filgueiras et al., 2019):
$$NDVI = \frac{\rho\text{NIR} - \rho\text{RED}}{\rho\text{NIR} + \rho\text{RED}} \qquad (1)$$
where $\rho$NIR and $\rho$RED are the spectral reflectances corresponding to the eighth (832.8–
832.9nm) and fourth (664.6– 664.9nm) Sentinel-2 MSI bands, respectively (Xu et al., 2021).
In general, NDVI values range from -1.0 to 1.0; the higher the NDVI value, the better the
conditions for vegetation development, and extremely low values indicate the presence of
water. Furthermore, the NDVI value above 0.4 indicates vegetated surfaces, and those between
0.25 and 0.40 signify soils with the presence of vegetation (Charrua et al., 2021). The vigor of
the vegetation increases as the NDVI values come closer to 1.00 (Rouse et al., 1974). Numerous
studies have established the NDVI threshold for vegetated land (e.g., Xu et al., 2021; Wong et

al., 2019; Liu et al., 2015; Eastman et al., 2013; Yang et al., 2012; Sobrino et al., 2004). Most researchers noted that the NDVI threshold value for vegetation cover typically ranges from 0.15-2.0 (Xu et al., 2021; Eastman et al., 2013; Sobrino et al., 2004). Therefore, the vegetated pixels (e.g., NDVI threshold > 0.20) present in pre and post-typhoon NDVI images have been used for vegetation severity analysis. The NDVI threshold is considered to reduce the effect of land cover change from the pre-typhoon (2018-08-01) to post-typhoon (2018-10-15) periods.

The degree of vegetation damage has been determined by comparing the NDVI values of the pre-and post-typhoon periods. Various researchers have frequently used the direct difference of NDVI to determine the damage severity caused by typhoons to naturally vegetated land (Wang and Xu, 2018; Konda et al., 2018). It has been calculated on a cell-by-cell basis by subtracting the pre-typhoon NDVI image from that of the post-typhoon in ArcGIS software using map algebra (Zhang et al., 2013; Cakir et al., 2006). The following equation is used to calculate the ΔNDVI (Wang and Xu, 2018),

$$\Delta NDVI = NDVI_{post-typhoon} - NDVI_{pre-typhoon} \qquad (2)$$

The difference in NDVI (i.e., ΔNDVI) illustrates the change in natural vegetation, while a negative ΔNDVI value indicates the damage inflicted by a typhoon to the vegetation cover (Xu et al., 2021).

The relative change in NDVI value has been used to investigate the geo-ecological impact on the forest area (Mishra et al., 2021b). The relative vegetation changes ($NDVI_r$) after Soulik have been determined by using the following Eq. (3) (Kumar et al., 2021),

$$NDVI_r = \frac{\Delta NDVI}{NDVI_{pre-typhoon}} \times 100 \qquad (3)$$

where the negative $NDVI_r$ value indicates vegetation loss caused by typhoons, and the positive $NDVI_r$ value shows vegetation gain. The $NDVI_r$ value has been classified into three categories corresponding to pixels with decreased, no change, or increased vegetation cover.

On the other hand, we analyze FVC in conjunction with NDVI, which provide additional insights into vegetation conditions and damage severity. Numerous researchers (Wang and Xu, 2018; Song et al., 2017; Bao et al., 2017; Chu et al., 2016; Amiri et al., 2009) used FVC to analyze vegetation damage, restoration, recovery, and inter-annual variability. In the present study, FVC was calculated before and after the typhoon using the derived NDVI data (Wang and Xu, 2018). It is expressed as a percentage and can range from 0 to 100%. The formula of FVC is as follows (Wang and Xu, 2018; Amiri et al., 2009; Carlson and Ripley,

1997),

$$FVC = [(NDVI - NDVI_m)/(NDVI_{max} - NDVI_m)]^2 \qquad (4)$$
where, $NDVI_m$ and $NDVI_{max}$ represent the $NDVI_{min}$ and $NDVI_{max}$ values calculated using
equation (1) (Zhang et al., 2021; Ge et al., 2018). The calculated FVC values vary between 0
and 1. After that, the FVC values were converted to percentages to fit the actual FVC
classification scheme (Wang and Xu, 2018), which consists of five classes: high (80-100%),
medium-high (60-80%), medium (40-60%), medium-low (20-40%), and low (0-20%). Further,
the difference in FVC values between the pre-and post-typhoon images was used to calculate
the extent of vegetation damage using the following equation,
$$\Delta FVC = FVC_{post-typhoon} - FVC_{pre-typhoon} \qquad (5)$$
where, $\Delta FVC$ denotes the difference between the pre and post-typhoon FVC. The $\Delta FVC$ value
represents alterations in vegetation conditions and damage intensity, while a negative value of
$\Delta FVC$ indicates the extent of damage caused by a typhoon to vegetation cover (Wang and Xu,

2018).


**3.2.2 Coastal landform classification and change analysis**
Typhoons have adversely affected the coastal landform and ecology of the south and
west coasts of the Korean peninsula every year. Therefore, a GIS-based coastal change model
has been developed to understand the morphodynamics of coastal landforms during typhoons.
In the present study, we considered four coastal landform classes, i.e., wetland, wetland
vegetation, land, and water, for the coastal morphodynamic analysis (Maiti and Bhattacharya,
2011). The method consists of two algorithms, i.e., (a) the ISODATA algorithm used to classify
the coastal landform with four main classes, i.e., water, wetland, wetland vegetation, and land,
and (b) the change detection technique used to quantify the short-term and medium-term
coastal changes. In this approach, we accentuate in-depth morphological changes and
emphasize minor changes along the Mokpo coast caused by typhoon Soulik.
The pre-and post-typhoon Sentinel-MSI images have been classified using the
unsupervised classification technique to distinguish among different coastal landforms of the
study region. This approach is used to determine which types of coastal landforms were
adversely affected by Typhoon Soulik and which of them have recovered more quickly than
others. ERDAS Image software has been used to run the unsupervised classification algorithm
(ERDAS, 1997). Based on the k-means algorithm, this technique reduces variability within
pixel clusters (Charrua et al., 2021; Aswatha et al., 2020; Bhowmik et al., 2013). Finally, pre-

and post-typhoon Sentinel-2 MSI images have been classified into four coastal landform classes: land, water, wetland, and wetland vegetation.

The accuracy assessment is a commonly used method to determine how closely the classified map matches the reference data (Congalton, 1991). In the present study, the classified data (i.e., coastal landforms maps) have been derived through an unsupervised classification technique, while 550 random samples collected from different parts of the Sentinel- 2MSI standard false-color image are considered reference data. Thereafter, a confusion matrix was developed based on the reference and classified data to evaluate accuracy statistics (Story and Congalton, 1986). The *kappa* coefficient ($k$) has been used to determine the quantitative accuracy of the classified map (Landis and Koch, 1977). The assessment is quantified using three different statistics: overall accuracy, producer accuracy, and user accuracy (Story and Congalton, 1986). The model's precision is classified into five categories based on the $k$ values: near perfect ($k > 0.8$), substantial ($0.6 < k < 0.8$), moderate ($0.4 < k < 0.6$), fire ($0.2 < k < 0.4$), and poor ($k < 0.2$) (Landis and Koch, 1977).

The land transformation model based on mutual spatial replacements has been applied during the post-classification stage, as shown in Figure 4. The classified coastal landform classes, such as land, wetland, wetland vegetation, and water, have been spatially replaced in order to create coastal-change units. For example, the coastal landform class of wetland vegetation in the pre-typhoon period replaced by water in the post-typhoon period indicates the change class of wetland vegetation replaced by water. A total of nine coastal-change classes have been derived, as illustrated in Figure 4(b).

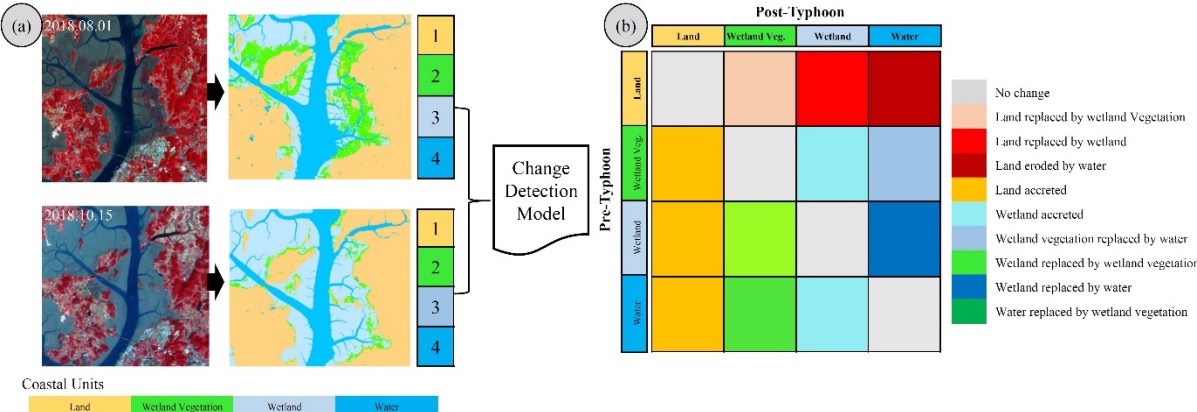

Figure 4. The coastal-change model exhibits spatial replacements among coastal landform classes.

### 3.2.3 Suspended sediment concentration modeling

The suspended sediment concentration (SSC) distribution in coastal regions is a significant indicator of changes in the marine environment caused by typhoon-induced storm surges, strong waves, and subsequent coastal flooding (Min et al., 2012; Gong and Shen, 2009). In a short period, a typhoon may drastically influence the water column structures (Souza et al., 2001), change the transport and deposition of sediment (Li et al., 2015), and affect the distribution of nutrients and biological production in the affected seas (Wang et al., 2016). Extreme storms or typhoons can modify suspended sediment distribution in coastal areas, which can significantly change marine habitats (Chau et al., 2021; Lu et al., 2018; Li and Li, 2016). Due to strong typhoon wind stress, the concentration of suspended particles in the seawater column and sediment resuspension may increase dozens of times before and after the event (Lu et al., 2018; Bian et al., 2017). Thus, typhoons significantly affect suspended sediment movement in the coastal region (Zhang et al., 2022; Li and Li, 2016; Goff et al., 2010). The spatiotemporal distribution of SSC can be impacted by variations in tidal phase, runoff, and wind speed (Tang et al., 2021). Furthermore, the resuspension of sediment can cause numerous problems in ocean engineering and change the region's ecology (Kim, 2010). The amount of material delivered to and adverted across the shelf by typhoons is considerably larger than that of winter storm systems (Dail et al., 2007). The southern and western part of the Korean peninsula is affected by an average of three typhoons annually passing through the Yellow Sea (KMA, 2018; Altman et al., 2013). Some studies on SSC distribution impacted by artificial construction along the coastal region of the Yellow Sea have been undertaken by several researchers (i.e., Lee et al., 2020; Eom et al., 2017; Min et al., 2012, 2014; Choi et al., 2014). However, the effects of typhoons on the sedimentary environment in the Mokpo coastal region have not yet been investigated. Therefore, it is imperative to carry out regional-scale SSC mapping and coastal modifications to reveal changes in the marine environment and sediment transport mechanisms over the typhoon period.

Remote sensing has long contributed to the advancement of water quality studies (Hossain et al., 2021). In the present study, we attempted to calculate both the qualitative and quantitative SSC in the inner-shelf region of the Mokpo coast using Sentinel-2B MSI data. The relative suspended sediment concentration has been calculated from pre- and post-typhoon Sentinel-2B MSI images using the NDSSI. NDSSI has been used in various water quality research (Kavan et al., 2022; Hossain et al., 2010). Further, many studies (Shahzad et al., 2018; Arisanty & Saputra, 2017) have successfully used Landsat and Sentinel-2 data to calculate

NDSSI. This index determines the relative concentration of suspended sediment, with values
ranging from -1 to 1, where -1 indicates the highest concentration and +1 indicates the lowest
(Hossain et al. 2010). The NDSSI has been calculated by using Eq. (6).
$$NDSSI = \frac{\rho Blue - \rho NIR}{\rho Blue + \rho NIR} \qquad (6)$$
where ρBlue and ρNIR represent the surface reflectances of Band 2 (492.1– 492.4 nm) and
Band 8 (832.8 – 833.0 nm) of Sentinel-2 MSI data, respectively. The NDSSI is based on the
observation that turbid waters reflect more in the NIR band but less in the visible band. The
negative NDSSI value represents that the reflectance of water in the NIR band is greater than
that in the blue band (Shahzad et al., 2018; Hossain et al., 2010). Therefore, the positive values
of NDSSI represent lower SSC or more transparent water, while a negative value indicates
higher SSC. The spatial patterns of relative SSC during the typhoon period have been
determined using the NDSSI.

On the other hand, the empirical model has also been used to quantify the suspended
sediment concentration before and after typhoon Soulik. This method is widely used for SSC
mapping and monitoring around the world (Eom et al., 2017; Hwang et al., 2016; Son et al.,
2014; Min et al., 2012; Lee et al., 2011; Choi et al., 2014). For this purpose, we reviewed the
existing relations between the in-situ SSC (SSC, $g/m^3$) and remote sensing reflectance ($R_r$)
developed by various researchers for the southern and western coasts of South Korea, as
illustrated in Table 2. In the present study, the SSC algorithm developed by Choi et al. (2014)
for the Mokpo coastal region based on the in-situ SSC and a spectral ratio of water reflectance
around 660nm has been used to quantify the SSC distribution. The atmospheric corrected
sentinel-2 MSI image (Red band) has been used to calculate the SSC.

Table 2. Relationship between the remote sensing reflectance ($R_r$) and suspended sediment
concentration (SSC, $g/m^3$).

| Authors | Relation | Region | Wavelength (nm) |
|---|---|---|---|
| Min et al. (2012) Min et al. (2006) | $Y=0.24e^{188.3x}$ | Saemangeum coastal area | 560nm |
| Choi et al. (2014) | $Y=1.545e^{179.53x}$ | Mokpo coastal area, Gyeonggi Bay | 660nm |
| Lee et al. (2011) | $Y=16.2064e^{15.3529x}$ | Gwangyang Bay and Yeosu Bay | 565nm |
| Choi et al. (2012) Lee et al. (2020) | $Y=1.7532e^{204.26x}$ | Yellow Sea | 660nm |
| Eom et al. (2017) | $Y=1.5119e^{179.85x}$ | Nakdong River | 660nm |

| Min et al. (2004) | $Y=0.99e^{199.9x}$ | Saemangeum | 560nm |


**3.2.4 Coastal erosion and accretion analysis**
The shorelines (i.e., land and water boundary) of the Mopko coast for short and medium
periods have been extracted using a semi-automatic technique (Maiti and Bhattacharya, 2009).
Here, we used the normalized difference water index (NDWI) and manual digitization
approach to separate the land and water boundary. The technique is widely used for dividing
land and water boundary (Santos et al., 2021; Dai et al., 2019). By using Sentinel-2 imagery,
NDWI can be achieved with the following formula (McFeeters, 1996),
$NDWI = \frac{\rho Green - \rho NIR}{\rho Green + \rho NIR}$        (7)
where ρGreen is the green band, and ρNIR is the near-infrared band of Sentinel-2 MSI data.
The extracted land and water boundary of the Mokpo region are then converted into
polygons, and the shoreline has been determined using ArcGIS software. The shoreline change
statistics have been calculated using the DSAS program (Thieler et al., 2009). The extracted
shoreline for pre-and post-typhoon periods has been merged, and a 10m interval transect
perpendicular to a baseline has been created (Santos et al., 2021). After that, the NSM method
was used to calculate the total shoreline movement (in meters) between the pre-and post-
typhoon shoreline positions of each transect (Kermani et al., 2016).
$NSM = sh_{post} - sh_{pre}$        (8)
where $sh_{post}$ and $sh_{pre}$ represent the post and pre-typhoon shoreline positions, respectively.
On the other hand, the back-shore surface area changes due to shoreline movement
(retreat/advance) over the typhoon period has also been calculated using the geo-statistical
analyst tool. Several researchers (Awad and El-Sayed, 2021; Deabes, 2017; Karmani et al.,
2016) have also previously mapped the surface changes of the backshore region. To create the
surface area-change map, we first generated two polygon layers based on the extracted
shoreline, one for the pre-and one for the post-typhoon periods. Next, we utilized the
Symmetrical Difference tool in ArcGIS software to compute the difference between these
polygon layers during the period affected by the typhoon. Finally, two feature classes have
been derived, one for erosion and another for accretion. In addition, the attribute table contained
in each zone illustrates the magnitude of spatial changes (amounts of erosion and accretion)
during the typhoon period.

**4. Result and Discussion**

**4.1 Vegetation damage severity mapping (VDSM) before and after Typhoon**

*4.1.1 VDSM based on the NDVI and FVC analysis*

The VDSM shows the degree of vegetation damage due to typhoons. The comparison of pre-and post-typhoon NDVI and FVC distribution shows a significant loss of vegetated land as the number of no-productivity and low-productivity pixels increases in the post-typhoon NDVI and FVC image.

Figure 5 depicts the spatial distribution of pre and post-typhoon NDVI images. Further, to determine the severity of vegetation damage, the pre-and post-typhoon NDVI image has been classified into six categories, namely non-vegetation (-1.0-0.0), low-vegetation (0.0-0.2), medium-low vegetation (0.2-0.4), medium vegetation (0.4-0.6), medium-high vegetation (0.6-0.8) and high vegetation (0.8-1.0). The pre and post-typhoon mean NDVI values were observed to be 0.159 and 0.143, respectively, indicating a mean NDVI value decline of 0.016 after the typhoon.

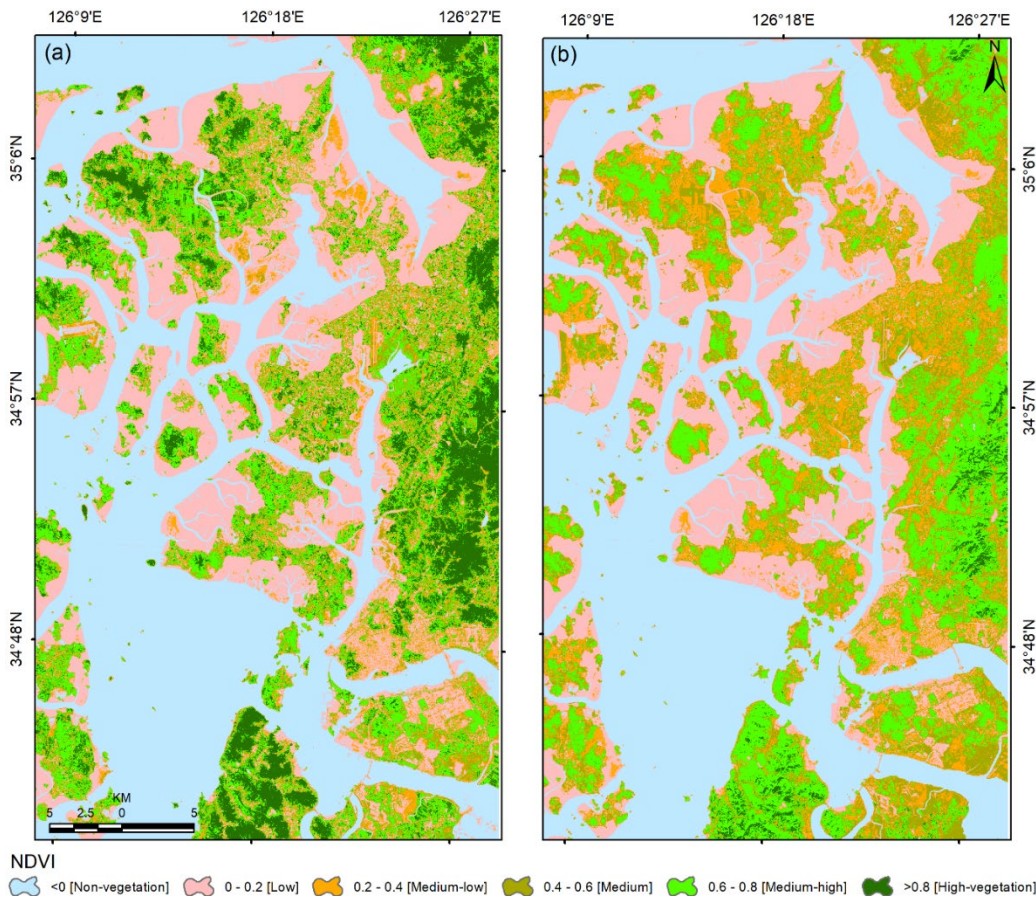

Figure 5. Status of vegetation greenness based on the NDVI data for the (a) pre-Soulik (01st August 2018) and post-Soulik (15th October 2018) period.

Table 3 depicts the area changes for each NDVI category over the typhoon period. It

has been observed that the high NDVI values (>0.8) have changed drastically after typhoon-
Soulik. The area changes in the low and non-vegetation categories along the Mokpo coastal
region revealed that the wetland (mudflat) had accreted after the typhoon. On the other hand,
the post-typhoon image was acquired two months after typhoon Soulik, which suggests that
the grasses and crops have recovered well. This recovery is reflected in Table 3 from medium-
low to medium-high NDVI levels.
Table 3. NDVI distribution over the study area before and after the typhoon.

| NDVI levels | Pre-typhoon (km$^2$) | Post-typhoon (km$^2$) | Change (km$^2$) |
|---|---|---|---|
| Non-vegetation (-1 to 0) | 673.7 | 647.6 | -26.2 |
| Low (0 to 0.2) | 430.4 | 415.2 | -15.2 |
| Medium-low (0.2 to 0.4) | 141.6 | 243.3 | 101.6 |
| Medium (0.4 to 0.6) | 132.5 | 225.3 | 92.8 |
| Medium-high (0.6 to 0.8) | 283.7 | 294.4 | 10.7 |
| High (0.8 to 1.0) | 183.6 | 19.8 | -163.8 |


On the other hand, the physical presence of vegetation has also been measured using
FVC analysis. In general, NDVI provides information on the health and productivity of
vegetation, while FVC provides information on the physical presence and distribution of
vegetation. Figure 6 depicts the pre- and post-typhoon FVC map of the Mopko coast. The area
of each FVC category is illustrated in Table 4. The results reveal that the typhoon caused a
substantial decrease in FVC in the area, with the average FVC reducing significantly from
33.43% to 23.64% after the typhoon. It was observed that the medium-high to high FVC area
decreased from 485.4 km$^2$ to 211.9 km$^2$, while the medium-to-low FVC area increased from
1359.8 km$^2$ to 1633.3 km$^2$. The high FVC vegetation category was more severely affected and
decreased considerably after the typhoon. These results indicate that the typhoon significantly
impacted the wetland vegetation in the region.

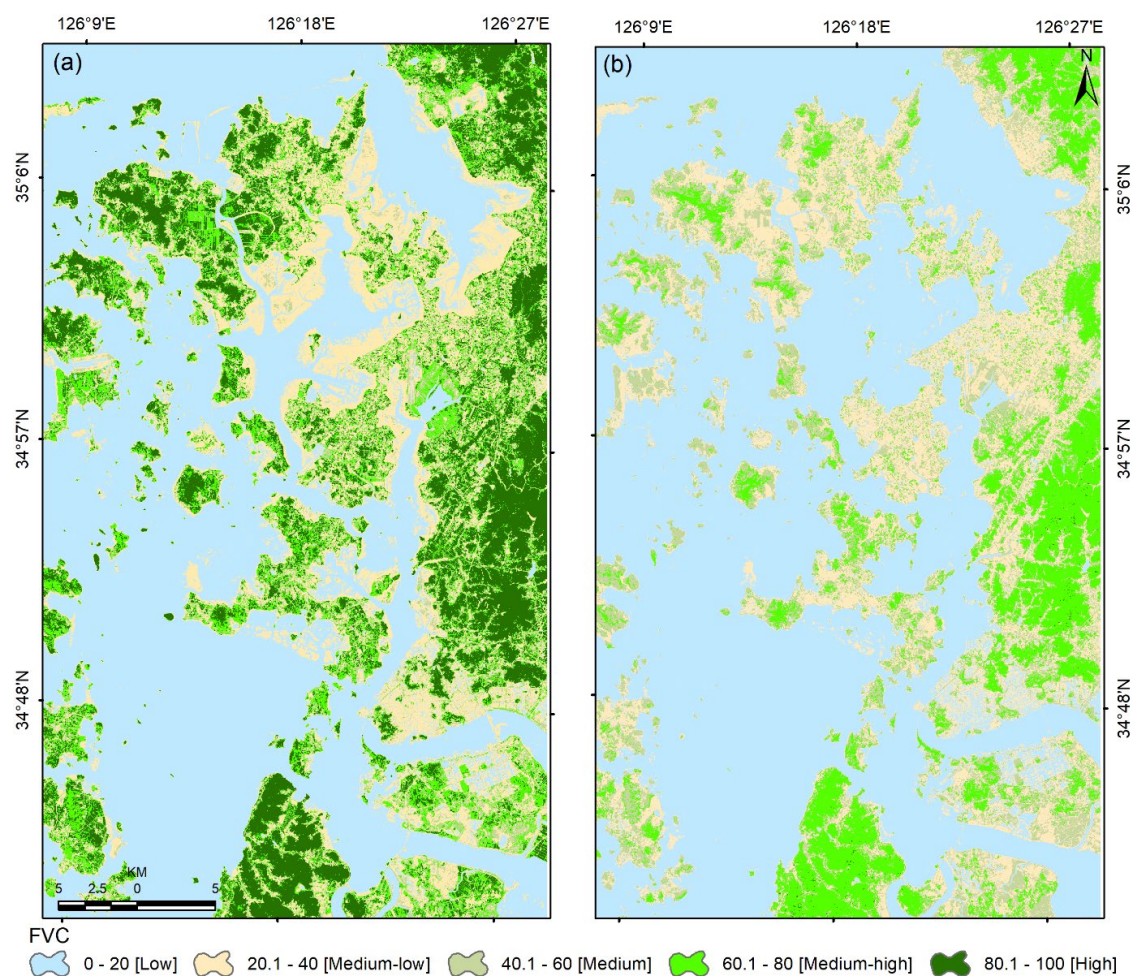

Figure 6. Status of vegetation based on the FVC analysis for the (a) pre-Soulik (01[st] August 2018) and post-Soulik (15[th] October 2018) period.

Table 4. Summary of FVC classes before and after the typhoon.

| FVC levels (%) | Pre-typhoon (km$^2$) | Post-typhoon (km$^2$) | Change (km$^2$) |
|---|---|---|---|
| Non-vegetation (<20) | 890.3 | 1053.3 | 162.943 |
| Medium-low (20-40) | 327.4 | 319.6 | -7.811 |
| Medium (40-60) | 142.4 | 260.6 | 118.205 |
| Medium-high (60-80) | 206.1 | 211.5 | 5.365 |
| High (80-100) | 279.4 | 0.7 | -278.671 |

In order to determine the damaged vegetation areas along the Mokpo coast, we compared pre-and post-typhoon NDVI images. A decrease in ΔNDVI is one of the most distinctive features of abrupt canopy modifications detectable by optical remote sensing (Xu et al., 2021). Thus, we can only determine vegetation deterioration from the two NDVI images. Subsequently, an NDVI threshold of 0.2 has been used to extract only vegetation features from the pre-and post-typhoon NDVI images. The threshold value has been manually adjusted to

achieve the highest accuracy of vegetation pixels. The extracted vegetated pixels have been
compared with reference samples randomly collected from the original high spatial resolution
images to determine the accuracy (Schneider, 2012; Xu et al., 2021). The two extracted
vegetation images obtained within six or seven weeks of typhoon Soulik's (i.e., before the
damaged vegetation had recovered) exhibits an overall accuracy of 95.7 % for pre-typhoon and
94.5% for the post-typhoon period.
Figure 7(a) depicts the spatial distribution of $\Delta$NDVI, where the highest $\Delta$NDVI
indicates a region with highly impacted vegetation areas. The negative $\Delta$NDVI is attributed to
about 26.7% of the total area (1845.60 km$^2$), which suggests that Typhoon Soulik affected
approximately 493.98 km$^2$ of vegetated land. The lowest $\Delta$NDVI value is -0.89, which
indicates either tree wind throws or a change in land surface cover from vegetation to build-up
land or other non-vegetation covers (Zhang et al., 2013). The results showed that wetland
vegetation and agricultural land experienced the most significant NDVI changes, with $\Delta$NDVI
values below-0.3. This suggests that these two types of land cover were severely affected by
typhoon Soulik.
On the other hand, Figure 7(b) displays the change map obtained from the difference in
FVC ($\Delta$FVC), which reveals areas of altered vegetation after the typhoon. The negative $\Delta$FVC
is attributed to about 32.07% of the total area, which suggests that Typhoon Soulik affected
approximately 591.89 km$^2$ of vegetated land. It has also been observed that the pure vegetation
pixels (i.e., NDVI>0.6 and FVC>60%) were drastically changed over the typhoon period. The
changed area determined for NDVI and FVC is -153.43 km$^2$ and -273.40 km$^2$, respectively
(Tables 3 & 4). The results obtained from both techniques indicate a significant decrease in
vegetation cover after the typhoon. The probable reason for the change is that Typhoon Soulik
made landfall close to Mokpo coastal region.

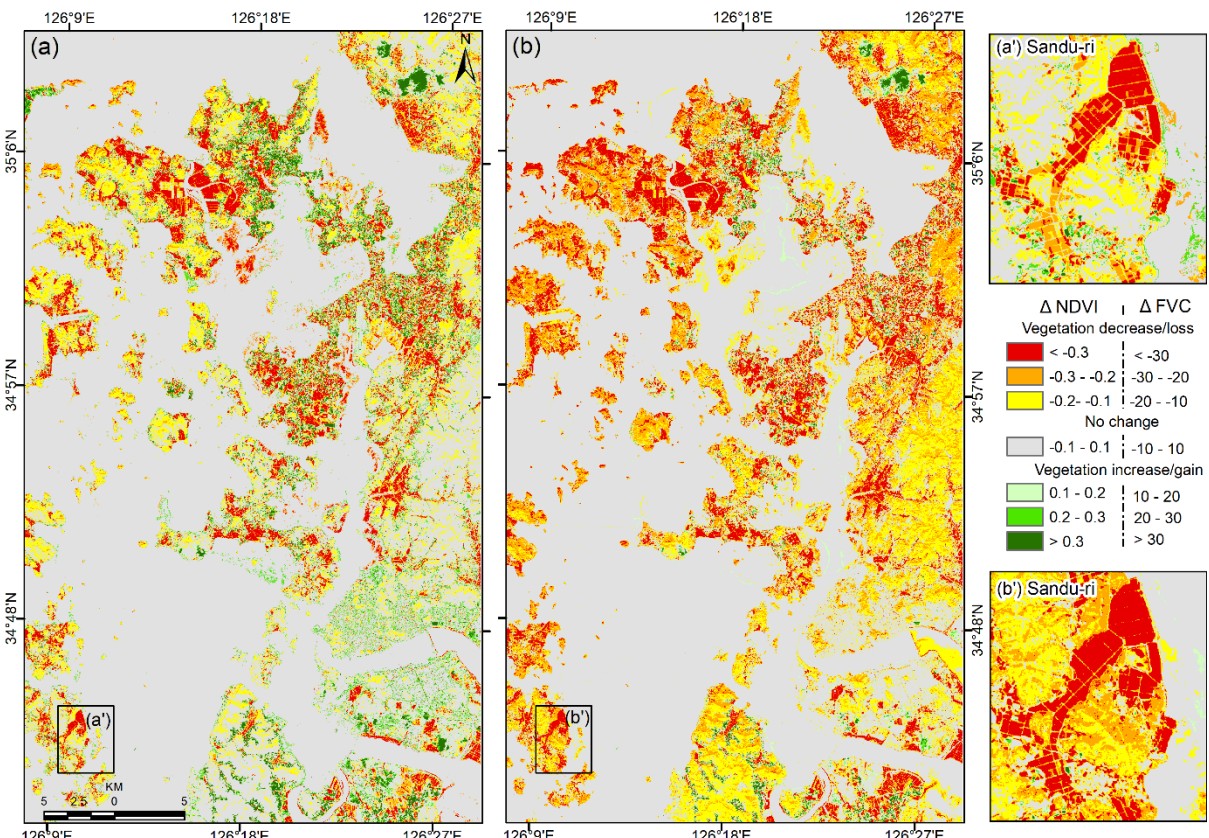


Figure 7. Vegetation change map of the Mokpo coastal region derived through two different
methods: (a) ΔNDVI and (b) ΔFVC, whereas zoom boxes show the vegetation
damage of Sandu-ri areas.

Figure 8 compares vegetation damage based on the number and percentage of the
decreased pixel of ΔNDVI and Δ FVC. It exhibits decreased pixels in different categories of
vegetation damage, ranging from low damage to extensive damage. The pixels showing the
most significant vegetation damage (i.e., ΔNDVI -0.2 to -0.5 and ΔFVC -20 to -50%) account
for about 30.9% and 61.5% of the total pixels, respectively. On the other hand, the pixels
showing extensive vegetation damage (i.e., ΔNDVI<-0.5 and ΔFVC<-50%) account for only
8.31% and 10.76% of the total pixels. It was observed that the dominant vegetation in the region
is wetland vegetation, which is mainly due to the prevalence of wetlands or mudflats in the
area. Therefore, the significant vegetation damage implies that wetland vegetation was most
severely impacted during typhoons.

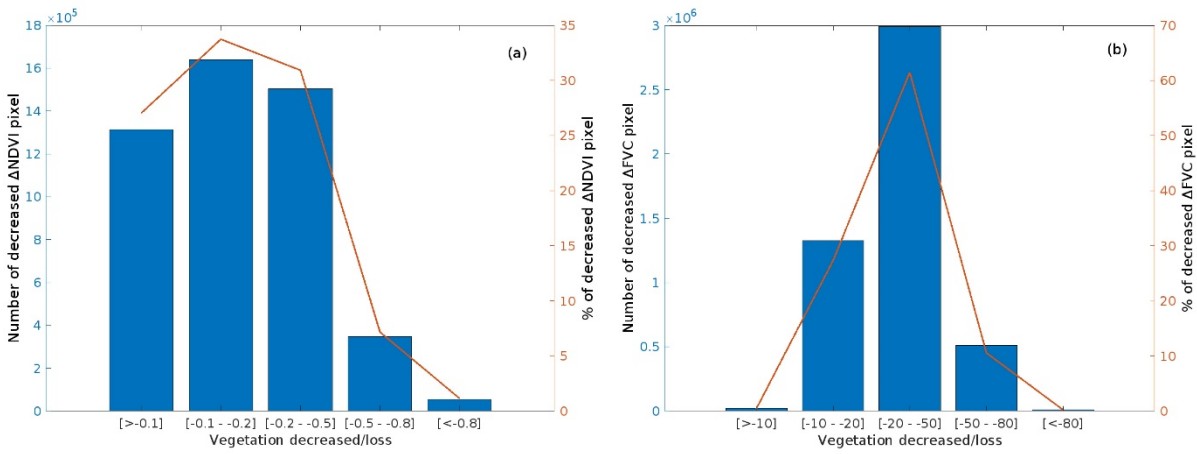


Figure 8. Comparison of vegetation damaged represented based on the number and percentage
of decreased pixels of (a) ΔNDVI and (b) Δ FVC.


The pre-and post-typhoon Sentinel-2 false-color images and the corresponding relative
change in $NDVI_r$ and ΔFVC values are presented in Figure 9. The standard FCC imagery (left
panel, Fig. 9) for pre and post-typhoon shows that $NDVI_r$ is more effective in detecting areas
of damaged vegetation compared to ΔFVC (right panel, Fig. 9). It was observed that the
typhoon-induced damaged vegetation area (i.e., pixels with $NDVI_r$ and ΔFVC of <-50%)
detected by $NDVI_r$ (106.5 km$^2$) was greater than that detected by ΔFVC (51.3 km$^2$). The
dissimilarity in the ability of $NDVI_r$ and ΔFVC to detect the destruction of vegetation caused
by the typhoon can be ascribed to the alteration in the color of the vegetation post-typhoon.
This change can be detected more accurately by NDVI compared to FVC because the
vegetation in the affected areas still existed, and there was not a significant reduction in
vegetation coverage after the event (Wang and Xu, 2018). Thus, NDVI is highly sensitive to
the health status of vegetation and a more appropriate approach for assessing the damage to
vegetation induced by the typhoon, while FVC is more representative of vegetation coverage
status (Wang and Xu, 2018; Jing et al., 2011). Consequently, the dramatic vegetation loss (<-
80%) that occurred in mostly wetland vegetation is detected mainly in $NDVI_r$. In addition,
moderate greenness loss has been identified in natural forests. Furthermore, the decrease of
$NDVI_r$ values from higher classes to lower classes indicates that the typhoon has severely
damaged the low-lying coastal regions and the wetland vegetation.

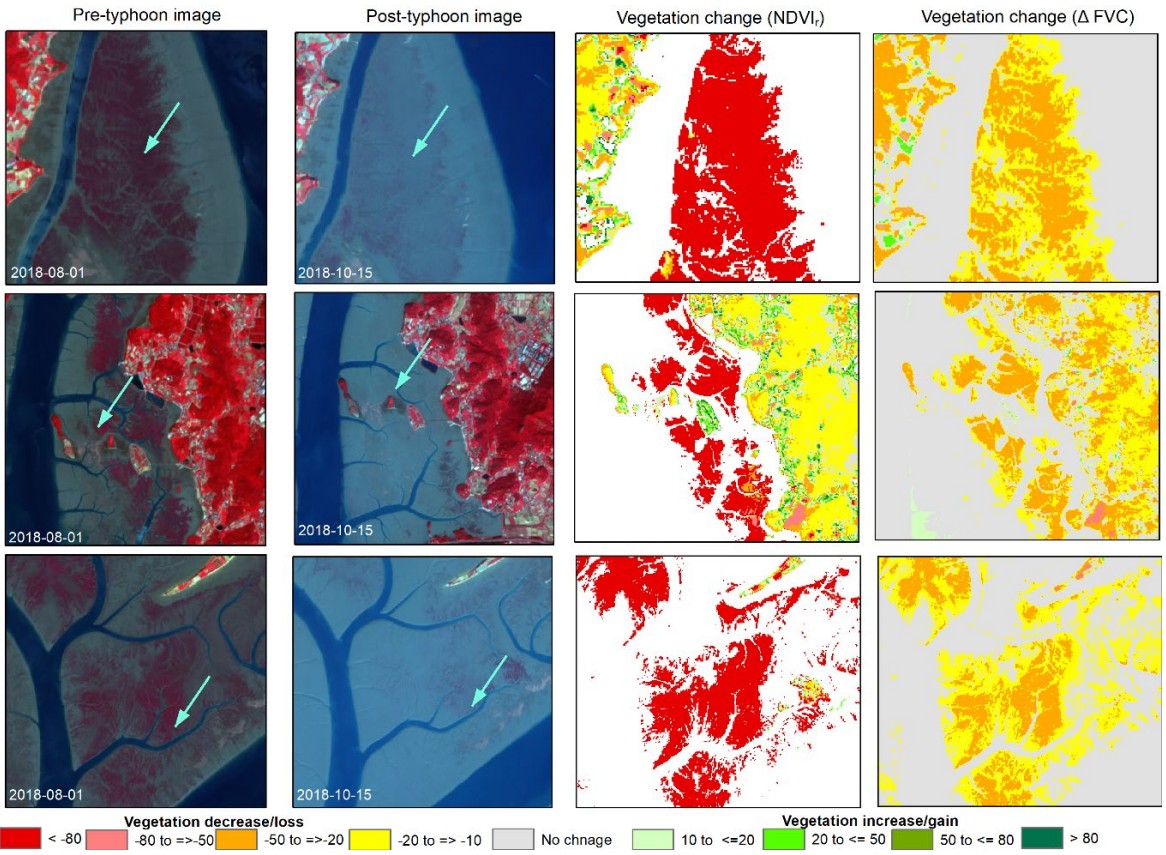

| | Vegetation decrease/loss | | | | | Vegetation increase/gain | | | |
|---|---|---|---|---|---|---|---|---|---|
| < -80 | -80 to =>-50 | -50 to =>-20 | -20 to => -10 | No chnage | 10 to <=20 | 20 to <= 50 | 50 to <= 80 | > 80 |

Figure 9. Sentinel-2 MSI standard false color composite images before and after Typhoon Soulik exhibit vegetation damage and the corresponding NDVI$_r$ and ΔFVC (Sentinel-2 MSI level 1C satellite images were downloaded from https://scihub. copernicus.eu/dhus/ ).

*4.1.2 Influence of topography on vegetation damage caused by Typhoon Soulik*

The affected area's topography can influence typhoons' impact on vegetation. The interaction between topography and typhoon-generated wind and rain can result in complex and varied patterns of damage across different landscapes (Abbas et al., 2020; Lu et al., 2020; Zhang et al., 2013). This can affect the severity and spatial patterns of vegetation damage. Therefore, the relationship between topography and damaged vegetation has also been established in the present study. For this purpose, high-resolution (5m×5m) DEM data provided by the NGII are used to calculate the region's topographic slope and explore the relationship between topography and typhoon-induced vegetation damage.

The Mopko coastal region showed an elevation range between 0 to 403 meters, as shown in Figure 1(b). It was observed that the number of trees damaged by Typhoon Soulik decreased as the elevation increased, as illustrated in Figure 10a. The highest number of damaged trees was observed in areas with an elevation of 50m or lower. This is likely due to the fact that these areas are predominantly covered by wetlands, which can be more vulnerable

to strong winds associated with typhoons Soulik. In general, low-lying areas may not have the
same natural windbreaks and barriers as higher elevations, which can exacerbate the impact of
the wind. In addition, low-elevated vegetation may have shallower root systems due to the less
stable soil conditions, making them more vulnerable to uprooting during heavy rainfall or
strong winds (Zhang et al., 2013; Lugo et al., 1983). A significant difference in the number of
decreased ΔNDVI and ΔFVC pixels was observed among different elevation ranges, and a
correlation analysis between the number of damaged pixels and elevations showed a negative
correlation (i.e., damaged pixels decreased with increasing elevation). The majority of
damaged pixels (76.37%) were observed at elevations between 0 and 50m, with a decrease to
13.5% between 51 and 100m. The vegetation exhibited a sharp decline at higher elevations, as
shown in Figure 10(a), with the proportion of pixels displaying negative ΔNDVI and ΔFVC
decreasing to 6.1% between 100 and 150m and decreasing to 0.02% between 350 and 403m.

On the other hand, Figure 10(b) illustrates the extent of damaged vegetation across
different slope ranges. It has been noted that there is a negative correlation between the slope
and the percentage of damaged vegetation pixels, indicating that the amount of vegetation
damage decreases with a higher slope. For instance, when the slope was between 0-5°,
approximately 47.63% of vegetation pixels were damaged. As the slope increased, the
percentage of damaged vegetation pixels decreased accordingly, with values of 18.15%,
15.01%, 10.71%, 7.74%, 0.73%, and 0.009% observed for slope ranges of 5-10°, 10-15°, 15-
20°, 20-30°, 30-40°, and greater than 40°, respectively.

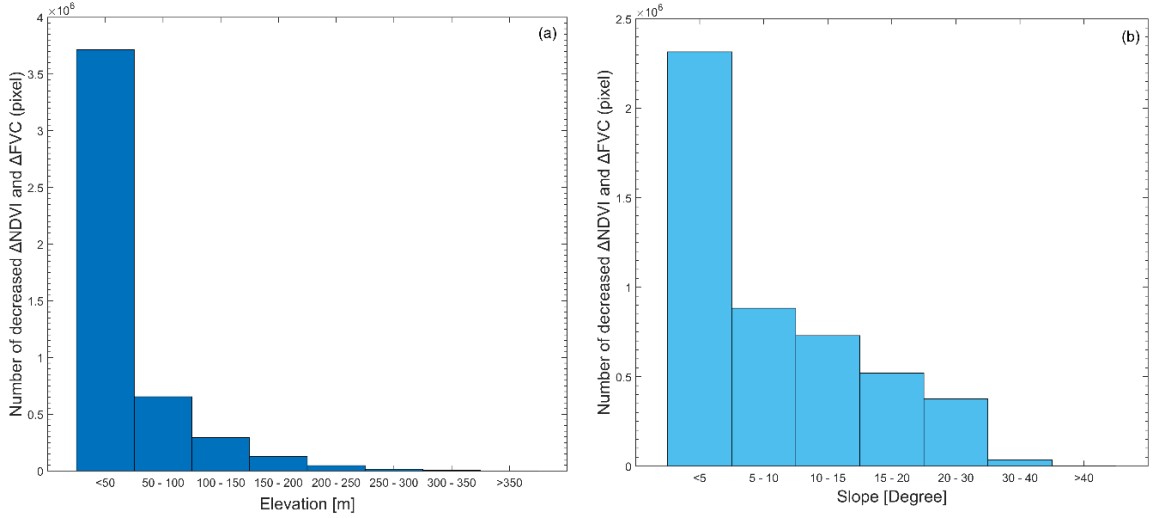


Figure 10. The relationship between topography and vegetation damaged due to typhoon
Soulik: (a) numbers of damaged vegetation at different elevation ranges, and (b)
numbers of damaged vegetation at different slope ranges.

**4.2 Coastal morphodynamics over the typhoon period**

To understand the coastal morphodynamics over the typhoon period (i.e., short-term), we classified the entire coastal region into four major coastal landform classes: land, wetland vegetation, wetland, and water (Fig. 11a-b). The accuracy and *kappa* coefficient of the classified maps exhibited a reasonable degree of consistency with the reference data, as illustrated in Table 5. The overall accuracy of the pre-and post-typhoon coastal landform maps was 86.5% and 84.3%, and *kappa* coefficients were 0.82 and 0.79, respectively. The results of the coastal landform classification showed a reduction in wetland vegetation over the typhoon period. Table 6 illustrates that before the typhoon, the area of the wetland vegetation class was 4.21% (77.63 km$^2$) of the total area of all categories (1845.60 km$^2$). However, after the hitting of the typhoon storm, the wetland vegetation area reduced to 1.08% (19.90 km$^2$), recording a degradation of 57.73 km$^2$ (-74.37%).

Table 5. Accuracy assessment of pre-and post-typhoon classified coastal units.

| Coastal Units | Description | Pre-typhoon | | Post-typhoon | |
|---|---|---|---|---|---|
| | | Producer Accuracy (%) | User Accuracy (%) | Producer Accuracy (%) | User Accuracy (%) |
| Land | Others Land use | 90.2 | 92.0 | 91.9 | 90.7 |
| Wetland vegetation | Wetland vegetation | 83.4 | 84.0 | 85.0 | 83.3 |
| Wetland | Mudflat/tidal flat | 81.4 | 84.7 | 77.1 | 74.0 |
| Water | Waterbody | 91.4 | 85.3 | 83.2 | 89.3 |
| Overall accuracy (%) | | 86.5 | | 84.3 | |
| *kappa* | | 0.82 | | 0.79 | |

The most remarkable gain was the wetland class after the typhoon. This is shown by an increase of wetlands from 258.14 km$^2$ to 334.97 km$^2$, i.e., an increase of 29.76% (76.83 km$^2$) during the short periods. Furthermore, the land class has increased by only 0.20% over the typhoon period, i.e., from 45.34% (before the typhoon) to 45.44% (after the typhoon). In addition, it has been noticed the waterbody decreased by 3.09% (20.78 km$^2$) after the typhoon. Thus, it can be inferred that most wetland vegetation and waterbody have been converted into wetlands, which caused the coastal deterioration.

Table 6. Area changes of the different coastal units during the pre-and post-typhoon periods in the Mokpo coast.

| Coastal Units | Area at pre-typhoon | Area at post-typhoon | Changed area |
|---|---|---|---|

|  | km$^2$ | % | km$^2$ | % | km$^2$ | % |
|---|---|---|---|---|---|---|
| Land | 836.87 | 45.34 | 838.55 | 45.44 | 1.68 | 0.20 |
| Wetland Vegetation | 77.63 | 4.21 | 19.90 | 1.08 | -57.73 | -74.37 |
| Wetland | 258.14 | 13.99 | 334.97 | 18.15 | 76.83 | 29.76 |
| Water | 672.95 | 36.46 | 652.18 | 35.34 | -20.78 | -3.09 |
| Total | 1845.60 | 100.00 | 1845.60 | 100.00 | --- | --- |

Thereafter, the coastal land transformation model was developed through mutual spatial replacements between coastal units. The land transformation model has identified the nine coastal-change units, as shown in Figure 11(c). The results show that the lowland coastal area drastically changed after the typhoon, where the majority of coastal classes have been transformed into wetlands or mudflats. Furthermore, approximately 5.61% of the land area has been replaced by wetlands and water, whereas 83.79% of the wetland area has accreted over the wetland vegetation and water due to the impact of typhoon Soulik (Table 7).

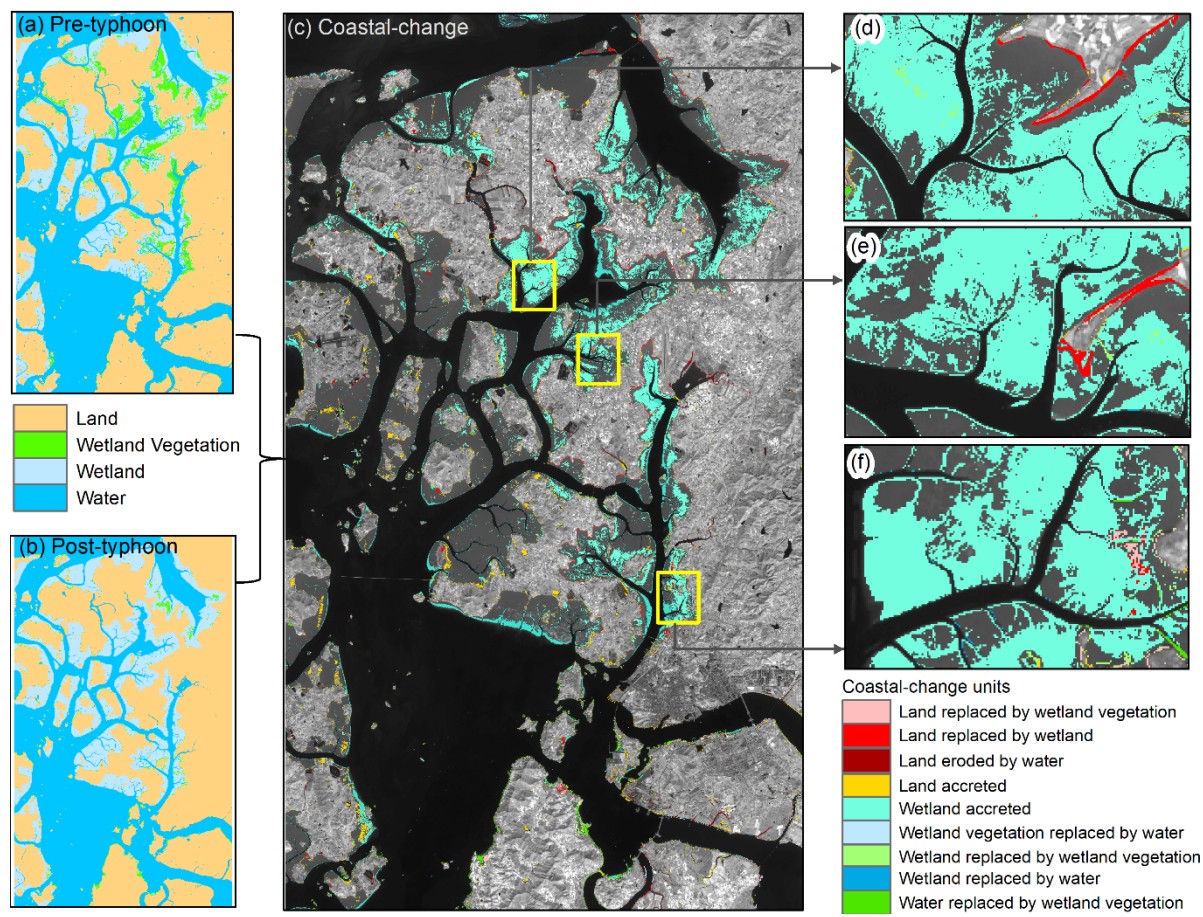

Figure 11. Spatial distribution of coastal-change units along the Mokpo coast due to typhoon Soulik: (a) pre-typhoon classified map, (b) post-typhoon classified map, and (c) coastal land transformation map. Subplots (d, e, and f) show the detailed coastal land

transformation.
Table 7. The details of coastal land transformation classes identify in short-period.

| Coastal land transformation | Area (km$^2$) | % |
|---|---|---|
| Land replaced by wetland vegetation | 4.59 | 3.94 |
| Land replaced by wetland | 4.41 | 3.79 |
| Land eroded by water | 2.12 | 1.82 |
| Land accreted | 12.88 | 11.06 |
| Wetland accreted | 83.79 | 71.97 |
| Wetland vegetation replaced by water | 2.47 | 2.12 |
| Wetland replaced by wetland vegetation | 1.59 | 1.36 |
| Wetland replaced by water | 1.76 | 1.52 |
| Water replaced by wetland vegetation | 2.82 | 2.42 |

**4.3 Sediment resuspension during the pre-and post-typhoon period**
The spatial distribution of relative suspended sediment concentration has been derived
through NDSSI for both before and after typhoon images (Fig. 12). Pre-typhoon SSC patterns
have been observed more SSC inside the creeks of the inner-shelf region of the Mokpo coast
as compared to the post-typhoon NDSSI image. However, it has been noted that the SSC has
significantly increased along the entire coast in the post-typhoon period (Fig. 12b). Therefore,
the spatial changes of relative SSC have been determined during the August (pre) and October
(post) periods, as depicted in Figure 12(c). In general, a flood always transports many
suspended materials and concentrates those materials on the upper surface of the water. After
the strong events, the flood-transported suspended material is deposited across the delta. A
similar phenomenon was observed in the post-typhoon period due to extensive rainfall, which
turned into a coastal flood.
On the other hand, it has been observed that the SSC gradually increased as the wind
speed increased from the pre to post-typhoon period. The increasing SSC amplitudes indicate
the rapid sediment erosion/resuspension over the storm passage. Furthermore, the amplitudes
of SSC variations were more visible in shallower water than in deeper water. The effect of
typhoons on the SSC variation along the Mokpo coast has been observed through ΔNDSSI
distribution (Fig. 12c). The negative ΔNDSSI values represent the increase of SSC due to
typhoon-induced strong wind and coastal flooding.

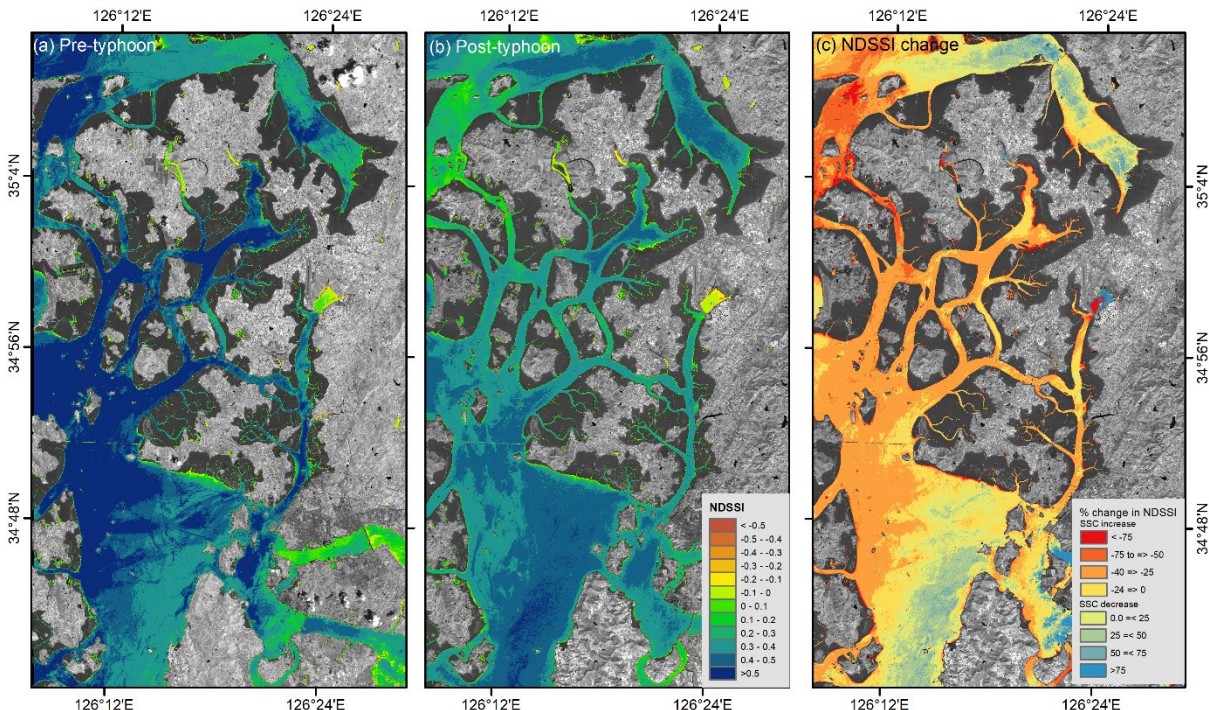

Figure 12. Relative SSC for (a) pre-typhoon and (b) post-typhoon period, while (c) represents the changes in the NDSSI.

Furthermore, a quantitative analysis of SSC has been performed based on the algorithm developed by Choi et al. (2014). During the pre-typhoon period, the SSC in the near shore waters was significantly higher than that of the offshore region (Fig. 13a). The post-typhoon image shows a sharp increase in the SSC distribution, indicating that Typhoon Soulik significantly impacted the SSC variation, with a maximum of >50 g/m$^3$ (Fig. 13c). In Figures 13(a) and (b), the spring-neap tidal influence broadly regulated the distribution and change of SSC throughout the shallow coastal water. The resuspension of SSC has been observed in the entire study region during the passage of Soulik. The pattern of relative SSC distribution (Fig. 12c) and the empirically derived SSC distribution (Fig. 13c) of pre-and post-typhoon are similar.

The outcomes showed that the storm surge and strong waves have considerably aided the sediment resuspension. Thus, the storm waves played an essential role in increasing bottom stress and stirring the seabed sediment (Gong and Shen, 2009). The transport of sediment during the storm adds another mechanism to the long-term morphological evolution of the Mokpo coast. This research revealed the profound significance of typhoons on inner shelf sedimentation along the coast.

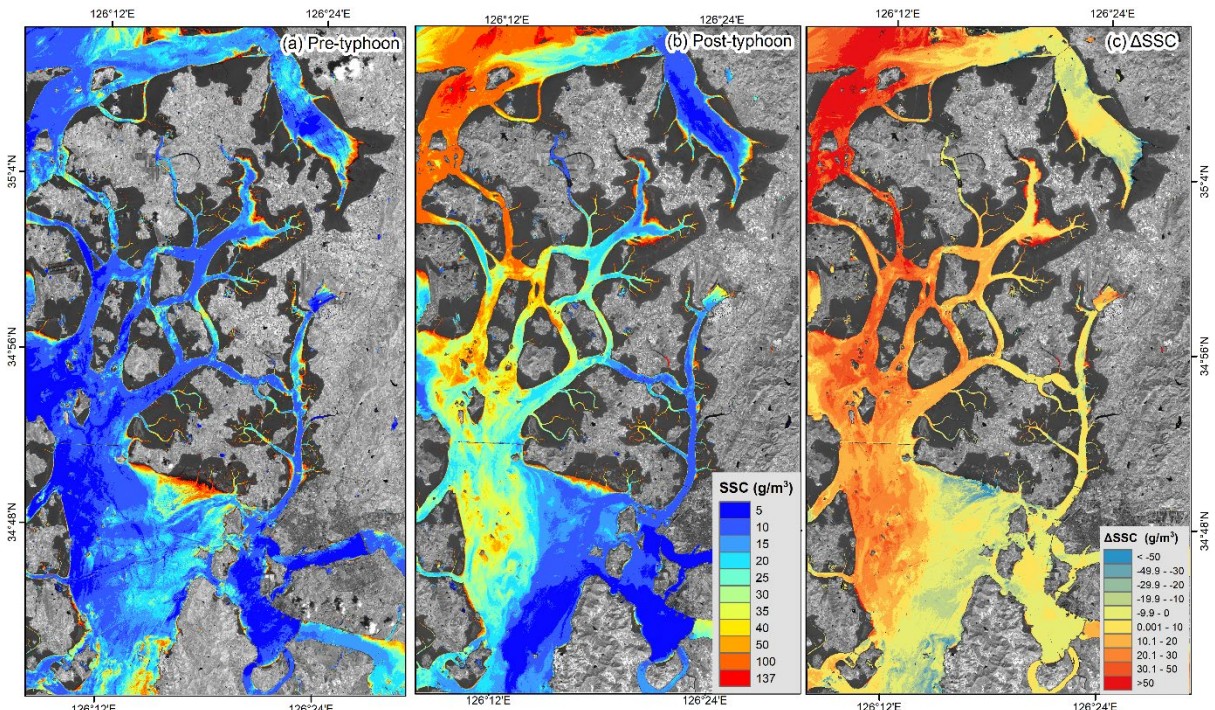

Figure 13. The simulated SSC distribution for the surface water of (a) pre-typhoon, (b) post-typhoon period, and (c) represents the spatial changes of SSC from pre- to post-typhoon.

**4.4 Impact on coastal erosion and deposition**

The impacts of the severe typhoon storm Soulik at a speed of 62 m/s on the coastline of Mokpo have been determined using the NSM method, considering 38313 transects (10m transect intervals) along the shoreline. Figure 14 shows the shoreline alteration in the entire Mokpo coastal region from the pre- to post-typhoon period (i.e., short-term), with an accretion of 87.5% transects and erosion of 12.5%. The mean deposition of 28.89m and a mean erosion of -8.29m were recorded (Table 8). The shoreline movement between 0-10m was recorded in the northern part of the coastal region. It has been observed that most transects experienced significant accretion; however, erosion has been observed in a few transects along the southern coastline (Fig. 14). The southern coast experienced sporadic landward movement of the shoreline. In contrast, the rest of the study region experienced significant seaward shoreline movement (Fig. 14 a-e).

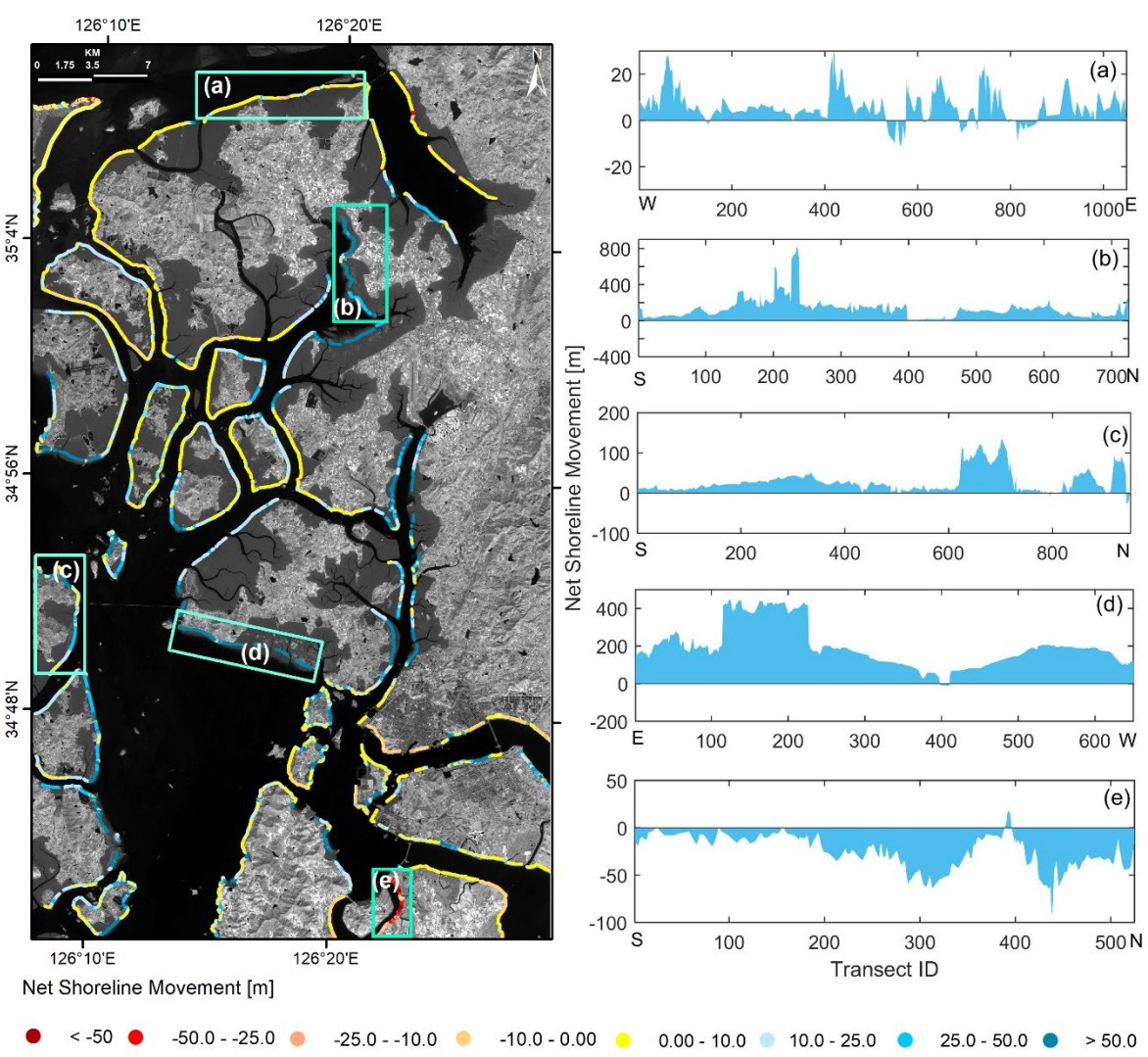


Figure 14. Short-term land water boundary changes (pre- to post-typhoon period) based on the
NSM method (left panel). Subplots (a-e) show the net movement of the shoreline at
different sites.


Table 8. Short-term (pre-post typhoon) shoreline change statistics based on the NSM model.

| NSM statistics | Summary |
|---|---|
| Total transects | 38313 |
| $NSM_{mean}$ | 24.24m |
| $NSM_{mean\ accretion}$ | 28.89 |
| $NSM_{mean\ erosion}$ | -8.29 |
| $NSM_{maximum\ accretion}$ | 812.54 |
| $NSM_{maximum\ erosion}$ | -131.72 |
| Total transect that records accretion | 34686 |
| Total transect that records erosion | 4955 |
| % of total transect that records accretion | 87.5 |
| % of total transect that records erosion | 12.5 |
| Overall pre to post-typhoon trend | Accretion |


The wind generated surface water currents that transported and dispersed erogenous

material to deep seas areas from pre- to post-typhoon. On the other hand, the coastal flooding
induced by the typhoon storm increased the sediment from the land to the near-shore region
(Figs. 12c & 13c). This allowed sediment to deposit on the wetland or beach areas. The coastal
land transformation map also revealed changes in shoreline shift-area as the wetland accreted
class.

The net surface area changes along the coastal region have been estimated and are

depicted in Figure 15. The total beach area increases and losses throughout the typhoon period
were 16.23 km$^2$ and 1.1 km$^2$, respectively (Fig. 15f). It was observed that typhoon Soulik
drastically increased the wetland (mudflat). These observations were also supported by other
proxies, as discussed above.

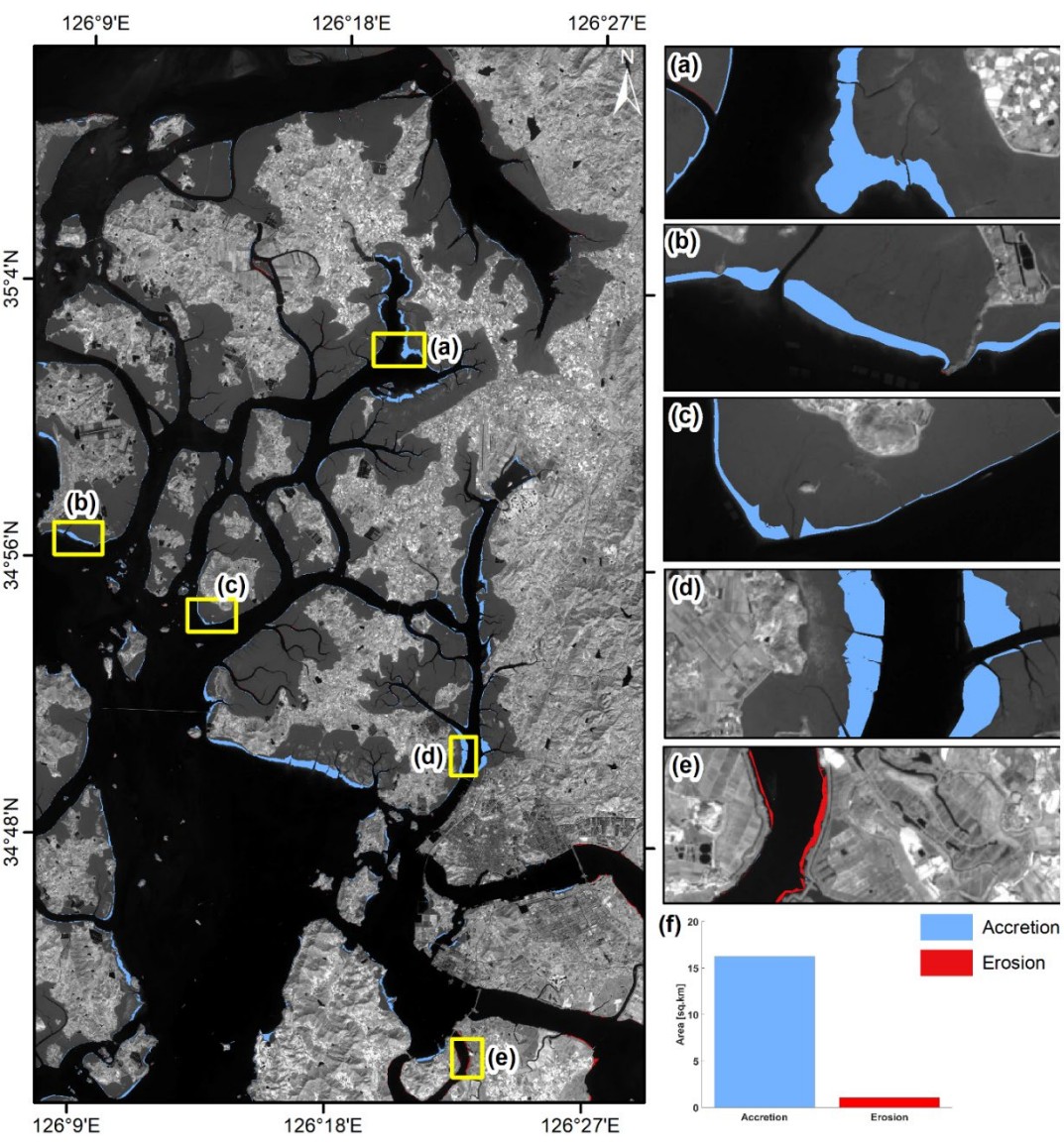


Figure 15. Short-term net surface area changes (i.e., erosion and accretion) due to typhoon Soulik along the Mokpo coast. Subplots (a-d) show extensive accretion, while erosion is shown in plot (e). The bar graph (f) represents the area changes from the pre to post-typhoon period.

**4.5 Coastal recovery status after typhoon Soulik**

The recovery status, i.e., medium-term coastal changes of the Mopko coastal region after typhoon Soulik has been analyzed using the NSM and coastal landform change model. For this purpose, another Sentinel-2 MSI level 1C satellite image was downloaded for October 2019 (one year after the typhoon), as listed in Table 1. After that, the coastal landform change model and NSM were performed based on the Sentinel-2 MSI images of October 2018 and 2019 (both images taken during the post-typhoon period) to understand the recovery status of the coastal morphometry. The coastal landform change model exhibits that the wetland vegetation increased drastically after one year of typhoon Soulik, as depicted in Figure 16. Table 9 indicates that approximately 16.52% of the land area has accreted over the wetland and water, whereas 39.71% of the wetland vegetation area has accreted over the wetland and water after the typhoon. Further, the outcome of the coastal recovery status was visually compared with the high-resolution aerial imagery obtained from the National Land Information Platform website (https://map.ngii.go.kr/) and showed strong agreement. Thus, the coastal landform change model successfully determined the longer-term recovery status in the topography and landforms of the Mopko coastal area after the typhoon.

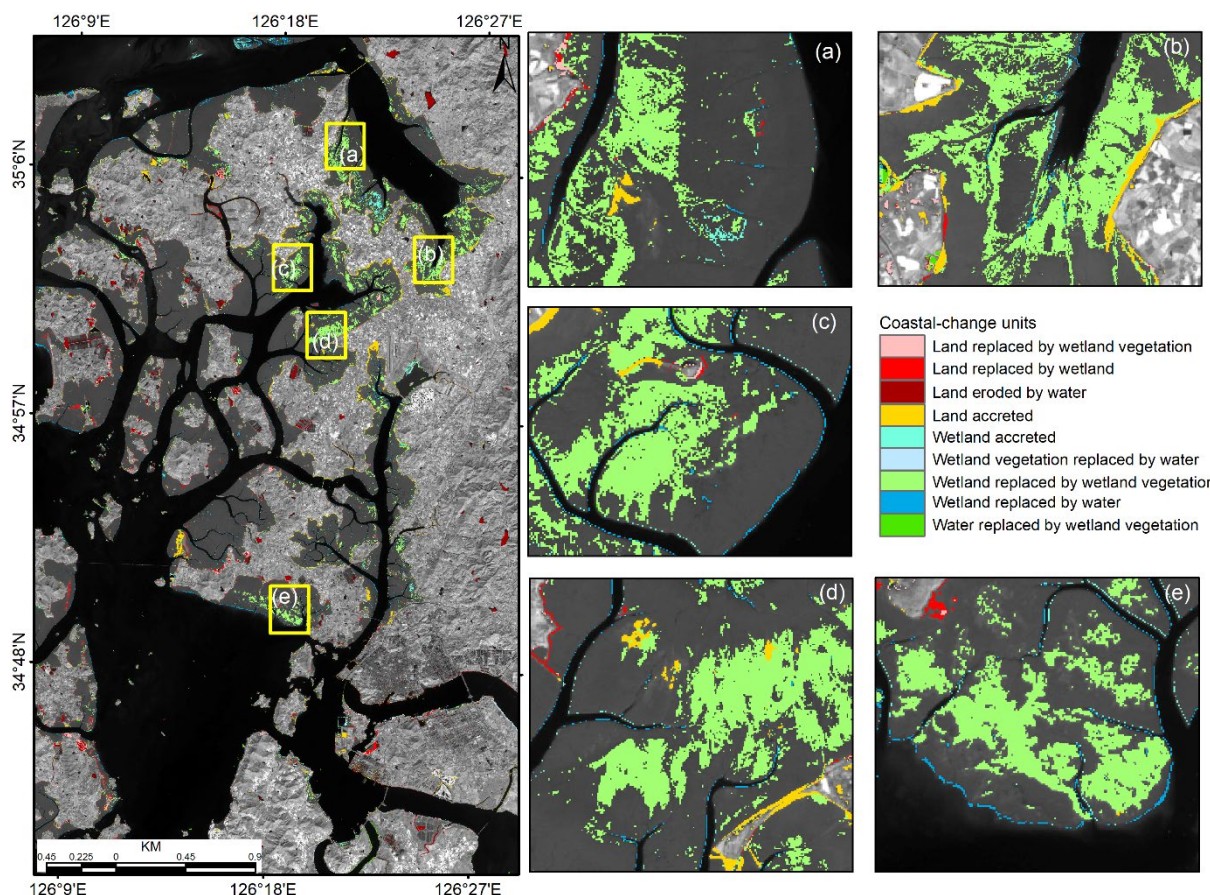

Figure 16. Recovery status of different coastal landforms after typhoon Soulik of Mokpo coastal region, whereas zoom boxes (a-e) show the increase of wetland vegetation at various sites.

Table 9. The details of medium-term coastal land transformation classes identify during the post-typhoon period.

| Coastal land transformation | Area (km$^2$) | % |
|---|---|---|
| Land replaced by wetland vegetation | 4.06 | 6.67 |
| Land replaced by wetland | 4.59 | 7.54 |
| Land eroded by water | 7.23 | 11.88 |
| Land accreted | 10.05 | 16.52 |
| Wetland accreted | 2.82 | 4.64 |
| Wetland vegetation replaced by water | 2.12 | 3.48 |
| Wetland replaced by wetland vegetation | 24.17 | 39.71 |
| Wetland replaced by water | 4.41 | 7.25 |
| Water replaced by wetland vegetation | 1.41 | 2.32 |

On the other hand, the medium-term effects of a typhoon on the shoreline have also been determined based on the NSM model. The results exhibit the extensive shoreline alteration in the entire Mokpo coastal region after one year of typhoon Soulik, with an accretion of 48.03% transects and erosion of 51.97%. The NSM statistics showed an average shoreline

movement of -1.08m, with a recorded mean erosion of -9.25 and deposition of 7.75m (Table

10). The overall erosion was recorded in response to typhoon Soulik even after one year along

the Mopko coastal region. This is due to the extensive damage to wetland vegetation during

the typhoon period (Table 7). In addition, it was observed that the wetland experience accretion

during the typhoon period, but it made the coastline vulnerable to erosion in the near future.

The natural native vegetation and wetland vegetation play a critical role in the shoreline

stability of the coastal region due to its anti-erosive nature. This phenomenon was evident in

the NSM statistics obtained during the post-typhoon period. Therefore, the use of these models

can help predict how the shoreline and adjacent coastal landforms will respond to typhoons,

identify vulnerable areas, and inform recovery efforts. This can enhance the area's resilience to

natural disasters and reduce the risk of future erosion and other environmental problems.

Table 10. Medium-term shoreline change statistics based on the NSM model.

| NSM statistics | Summary |
|---|---|
| Total transects | 38313 |
| $NSM_{mean}$ | -1.08m |
| $NSM_{mean\ accretion}$ | 7.75 |
| $NSM_{mean\ erosion}$ | -9.25 |
| $NSM_{maximum\ accretion}$ | 44.76 |
| $NSM_{maximum\ erosion}$ | -121.14 |
| Total transect that records accretion | 18400 |
| Total transect that records erosion | 19913 |
| % of total transect that records accretion | 51.97 |
| % of total transect that records erosion | 48.03 |
| Overall pre to post-typhoon trend | Erosion |

## 5. Conclusion

The objectives of this study were to assess the impact of typhoon Soulik on the coastal ecology,

landform, erosion/accretion, suspended sediment movement and associated coastal changes

along the Mokpo coast. This research developed an integrated approach for identifying coastal

dynamics impacted by typhoons and determining damage severity. The coastline movement,

coastal morphodynamics and quantified severity of vegetation damage from the pre- to post-

typhoon period have been determined based on the Sentinel-2 MSI images. NDVI and FVC

have been used to assess the severity of damage caused by typhoon Soulik on the vegetation.

The results showed that about 493.9 km$^2$ (26.7%) of vegetation had been affected in the Mokpo

coastal region. Further, it was observed that 6.1% (112.4 km$^2$) of vegetated areas in low coastal

land were severely damaged. The land transformation model exhibited that the 'wetland'

replaced most of the 'wetland-vegetated land' in the post-typhoon period. Also, it has been
found that more aggregated vegetation regions were less susceptible to damage.

The SSC of the Mokpo coastal region is higher in the post-typhoon period compared to
pre-typhoon time. The SSC variation influenced the coastal accretion and changed the deltaic
islands. The NDSSI and empirical-based SSC distribution of pre- and post-typhoon images
exhibit sedimentation drastically increased after the typhoon. The land accretion process also
dominated during the pre- to post-typhoon period. The wetlands and water have replaced
approximately 9.77% of the land area. On the other hand, 65.52% of the wetland area has
accreted over the wetland vegetation and water. Shoreline change analysis is also performed to
understand erosion and accretion in coastal regions. Typhoon Soulik accelerated shoreline
movement, affecting the local environmental condition, biodiversity imbalance, and aerial
change. In addition, 87.35% of shoreline transects experienced seaward migration over the
typhoon period. The wetland experience accretion in a shorter period, but it makes the coastline
vulnerable to erosion in the near future because the natural native vegetation and wetland
vegetation are crucial factors in shoreline stability of the coastal region due to its anti-erosive
nature. This phenomenon was evident in the NSM and coastal landforms change model
obtained in the medium-term analysis. However, more high-resolution, cloud-free multi-
temporal images and in-situ observations are required to better understand the medium to long-
term typhoon-induced morphodynamics of the coastal region. It can be concluded that the
Mokpo coastal ecosystem has been devastated by this extreme event. Although the observed
changes are not alarming, shoreline protection measures still need to be addressed, especially
the reforestation in wetland or mudflat regions. The outputs of the present study are needed to
better understand the sediment transport process and estuary changes during the pre-and post-
typhoon period. It can also be used to develop appropriate strategies to protect natural
ecosystems and post-disaster rehabilitation.

**Acknowledgments**
This paper was supported by research funds for newly appointed professors of Gangneung-
Wonju National University in 2021. The authors are thankful to the European Space Agency
(ESA) for providing free satellite images. The authors would like to thank the esteemed
reviewers for their valuable comments and suggestions that helped improve the manuscript.

**Funding**

This work was supported by the National Research Foundation of Korea (NRF) grant funded by the Korea government (NRF-2021R1C1C2003316) and Basic Science Research Program through the National Research Foundation of Korea (NRF) funded by the Ministry of Education (2021R1A6A1A03044326).

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
