# Peer review of "South Korea based on a geospatial approach"

_Natural Hazards and Earth System Sciences, 2022_

## Author Comment (AC1)

My co-authors and I would like to express our gratitude to the reviewers for their constructive feedback and suggestions for strengthening our research. The changes we have made to the attached file in response to such feedback and suggestions have been highlighted in blue to facilitate their identification. I would also like to offer my apologies for the length of time it took us to prepare this response. We also record our deep appreciation for the efficient handling of the manuscript.

**Response to Reviewer#1**

**General remarks**

I read the manuscript with great interest. Authors have investigated the impact of typhoon Soulik on the coastal ecology, landform, erosion/accretion, suspended sediment movement and associated coastal changes along the Mokpo coast. This research developed an integrated approach for identifying coastal dynamics impacted by typhoons and determining damage severity. Approach and analyses support to derive their conclusions.

The content is interesting for NHESS readers. Overall, the paper is well structured, with results being presented in a clear and organized manner. I have only a few comments and suggestions for improvements.

Thank you for reviewing our manuscript and suggesting that the subject of the manuscript is indeed of interest to NHESS. We considered your suggestions in the revised version of the manuscript, which has undoubtedly improved the contents and structure of the manuscript. Please find detailed responses to your comments.

**Comment 1:** Sections 3.3 and 3.4 should be discussed under section 3.2, i.e., Typhoon-induced coastal dynamic modeling. Accordingly, subsections should be renumbered and rearranged.

Response: Thank you for your insightful review. In the revised manuscript, sections 3.3 and 3.4 are discussed under section 3.2, and subsections have been renumbered and rearranged accordingly.

**Comment 2:** Figure 3 and Table 2 contain similar information. It is therefore recommended that Authors keep only one piece of information.

Response: Thank you for your insightful comment. We agreed with the reviewer's suggestion, and Figure 3 has been kept in the revised manuscript.

**Comment 3:** NDVI and FVC (Fractional vegetation coverage) are frequently used vegetation metrics for assessing land-surface vegetation conditions. Therefore, the use of

NDVI is reasonable for vegetation damage severity mapping. I would expect that Authors should analyze the FVC and compare it to NDVI-derived damaged severity. You are referred to go through the following paper: https://doi.org/10.1007/s11069-018-3351-7.

Response: This is a really interesting point raised by the reviewer. We agreed with the reviewer's suggestion and analyzed FVC in conjunction with NDVI, providing additional insights into vegetation conditions and damage severity. Subsequently, we compared the severity of vegetation damage obtained from both models (i.e., NDVI and FVC). Accordingly, sections 3.2.1 and 4.1.1 have been updated in the revised manuscript as,

**3.2.1 Analyses of coastal vegetation loss and disturbance**

[revised manuscript text omitted]

**Comment 4:** It would be better to explain the influence of topography on vegetation damage caused by Typhoon Soulik.

Response: Thank you for your insightful review. The affected area's topography can influence typhoons' impact on vegetation. The interaction between topography and typhoon-generated wind and rain can result in complex and varied patterns of damage across different landscapes (Abbas et al., 2020; Lu et al., 2020; Zhang et al., 2013). This affect the severity and spatial patterns of vegetation damage. Therefore, the relationship between topography and damaged vegetation has also been established in the present study. For this purpose, high-resolution (5m×5m) DEM data provided by the NGII are used to calculate the region's topographic slope and explore the relationship between topography and typhoon-induced vegetation damage.

It was observed that the elevation varies from 0 to 403 m in the Mopko coastal region, as depicted in Figure 1(b), and the number of trees damaged by Typhoon Soulik showed a decreasing trend at higher elevations (Fig 10a). The highest number of damaged trees was

observed in areas with an elevation of 50m or lower. This is likely due to the fact that these areas are predominantly covered by wetlands, which can be more vulnerable to strong winds associated with typhoons Soulik. In general, low-lying areas may not have the same natural windbreaks and barriers as higher elevations, which can exacerbate the impact of the wind. In addition, low-elevated vegetation may have shallower root systems due to the less stable soil conditions, making them more vulnerable to uprooting during heavy rainfall or strong winds (Zhang et al., 2013; Lugo et al., 1983). A significant difference in the number of decreased ΔNDVI and ΔFVC pixels was observed among different elevation ranges, and a correlation analysis between the number of damaged pixels and elevations showed a negative correlation (i.e., damaged pixels decreased with increasing elevation). The majority of damaged pixels (76.37%) were observed at elevations between 0 and 50m, with a decrease to 13.5% between 51 and 100m. Vegetation decreased rapidly at higher elevations, with the percentage of pixels with negative ΔNDVI and ΔFVC decreasing to 6.1% between 100 and 150m and decreasing to 0.02% between 350 and 403m, as depicted in Figure 10(a).

On the other hand, Figure 10(b) illustrates the extent of damaged vegetation across different slope ranges. It has been noted that there is a negative correlation between the slope and the percentage of damaged vegetation pixels, indicating that the amount of vegetation damage decreases with a higher slope. For instance, when the slope was between 0-5°, approximately 47.63% of vegetation pixels were damaged. As the slope increased, the percentage of damaged vegetation pixels decreased accordingly, with values of 18.15%, 15.01%, 10.71%, 7.74%, 0.73%, and 0.009% observed for slope ranges of 5-10°, 10-15°, 15-20°, 20-30°, 30-40°, and greater than 40°, respectively.

[Figure]

Figure 10. The relationship between topography and vegetation damaged due to typhoon Soulik: (a) numbers of damaged vegetation at different elevation ranges, and (b) numbers of damaged vegetation at different slope ranges.

**Comment 5:** A statistical summary of the shoreline change based on the NSM model should be presented in a tabular format.

Response: Thank you for your insightful comments. As suggested, the summary of shoreline change statistics based on the NSM model has been incorporated in the revised manuscript.

Table 8: Pre-post typhoon shoreline change statistics based on the NSM model.

| NSM statistics | Summary |
|---|---|
| Total transects | 38313 |
| $NSM_{mean}$ | 24.24m |
| $NSM_{mean\ accretion}$ | 28.89 |
| $NSM_{mean\ erosion}$ | -8.29 |
| $NSM_{maximum\ accretion}$ | 812.54 |
| $NSM_{maximum\ erosion}$ | -131.72 |
| Total transect that records accretion | 34686 |
| Total transect that records erosion | 4955 |
| % of total transect that records accretion | 87.5 |
| % of total transect that records erosion | 12.5 |
| Overall pre to post-typhoon trend | Accretion |

**Comment 6:** Line 66: The year of the reference in line 66 (Charrua et al., 2020) should be checked.

Response: Thank you for the comment. We have updated the text in the revised manuscript.

**Comment 7:** Line 112: The year of the reference in line 112 (Kwon et al., 2019) should be checked.

Response: Thank you for the comment. We have updated the text in the revised manuscript.

**Comment 8:** Line 139: The year of the reference in line 139 (Ryang et al., 2018) should be checked.

Response: Thank you for the comment. We have updated the text in the revised manuscript.

**Comment 9:** Line 306: The year of the reference in line 306 (Eom et al., 2016) should be checked.

Response: Thank you for the comment. We have updated the text in the revised manuscript.

**Comment 10:** Lines 335 and 342. Check the abbreviation of remote sensing reflectance.

Response: Thank you for the comment. The remote sensing reflectance ($R_r$) abbreviation has been checked and updated in the revised manuscript.

**Comment 11:** Line 461: The unit of measurement in Tables 6 and 7 should be standardized. Choose between sq km or km².

Response: Thank you for the comment. The unit of measurement (km²) in Tables 6 and 7 has been updated in the revised manuscript.

[revised manuscript text omitted]

---

## Author Comment (AC2)

My co-authors and I would like to express our gratitude to the reviewers for their constructive feedback and suggestions for strengthening our research. The changes we have made to the attached file in response to such feedback and suggestions have been highlighted in blue to facilitate their identification. I would also like to offer my apologies for the length of time it took us to prepare this response. We also record our deep appreciation for the efficient handling of the manuscript.

**Response to Reviewer#2**

**General remarks**

I found the article interesting, I think it makes important contribution in terms of disaster management caused by coastal erosion. In this study, the results were mapped using various models and index to analyze shoreline and coastal morphodynamics according to typhoons. It has been observed that typhoon-induced suspended sediment concentration influences shoreline and coastal morphology. This paper contributes to understanding natural disasters and their consequences in terms of scientific significance. I have a few comments (general and specific comments) and suggestions for improvements.

We greatly appreciate the critical review of the manuscript and the constructive suggestions put forth by the reviewer that will help improve the quality of the manuscript. We have responded point by point to all the comments and suggestions raised by Reviwer#2 as follows:

**Comment 1:** Figure 1(a) Is there a reason for showing the population above the basemap? If so, please comment on the difference between the color of the basemap in Figure 1(a) and the color of the basemap in Figure 1(b).

Response: Figure 1(a) is intended to illustrate the population density of the affected area, which is an important factor in understanding the impact of the typhoon on the affected region. On the other hand, the color of the basemap in Figure 1(a) represents the true color image (retrieved from ESRI World Imagery basemap), whereas the color of the basemap in Figure 1(b) represents the post-typhoon standard false-color composite image of the Mokpo coastal region (Sentinel-2 MSI data downloaded from https://scihub.copernicus.eu/dhus/). Both images (Figs. 1(a) and 1(b)) represent the extensive tidal flat region of the Mokpo coast. However, in the revised manuscript, we updated Figure 1 with more scientific exposition, such as province-wise recorded damage and loss distribution (Member Report, 2018), topography variation of the region (NGII, 2018), and variation of significant wave height and wind speed from August 20 to 25, 2018 recorded by Chilbaldo Buoy Station (located near the landfall area) during the

typhoon Soulik passage.

[Figure]

Figure 1. (a) Typhoon Soulik passage passed through the Mokpo coastal region on 23[rd] August 2018 (Typhon track data were downloaded from https://www.ncdc.noaa.gov/ibtracs/), while the background shades represent province-wise recorded damaged/loss distribution reported by Member Report (2018), (b) Topography variation of the Mokpo coastal region (elevation data acquired from NGII (2018), https://www.ngii.go.kr/, and bathymetry data downloaded from GMRT, https://www.gmrt.org), and (c) Variation of significant wave height and wind speed from August 20 to 25, 2018 recorded by Chilbaldo Buoy Station (located near the landfall area) during the typhoon Soulik (Data source: http://wink.kiost.ac.kr/map/map.do# ).

**Comment 2:** It would be better to add images to better understand the data in 3.1 Data Sources.

Response: As suggested, the pre-and post-typhoon standard false color composite images were incorporated in section 3.1 in the revised manuscript as,

[Figure]

Figure 2. Pre (a) and post-typhoon (b) standard false color composite of reflectance image of the Mokpo coastal region (Sentinel-2 MSI level 1C satellite images are downloaded from https://scihub.copernicus.eu/dhus/). The arrows indicate extensive vegetation damage due to Typhoon Soulik.

**Comment 3:** Short-term erosion caused by typhoons should be considered for recovery. It is necessary to confirm that the models (net shoreline movement (NSM) and coastal landform change) can predict the recovery of the shoreline and topography after a typhoon. and confidence in the model utilized (comparison with monitoring results, etc.) should also be mentioned.

Response: Thank you for your insightful comment. The recovery status of the Mopko coastal region after typhoon Soulik has been analyzed using the NSM and coastal landform change model. For this purpose, another Sentinel-2 MSI level 1C satellite image was downloaded for October 2019 (one year after the typhoon), as listed in Table 1. After that, the coastal landform change model and NSM were performed based on the Sentinel-2 MSI images of October 2018 and 2019 (both images taken during the post-typhoon period) to understand the recovery status of the coastal morphometry.

The coastal landform change model exhibits that the wetland vegetation increased drastically after one year of typhoon Soulik, as depicted in Figure 16. Table 9 indicates that approximately 16.52% of the land area has accreted over the wetland and water, whereas 39.71% of the wetland vegetation area has accreted over the wetland and water after the typhoon. Further, the outcome of the coastal recovery status was visually compared with the high-resolution aerial imagery downloaded from the National Land Information Platform web portal (https://map.ngii.go.kr/), indicating good consistency. Thus, the coastal landform change model successfully determined the longer-term recovery status in the topography and landforms of the Mopko coastal area after the typhoon.

[Figure]

Figure 16. Recovery status of different coastal landforms after typhoon Soulik of Mokpo coastal region, whereas zoom boxes (a-e) show the increase of wetland vegetation at various sites.

Table 9. The details of coastal land transformation classes identify during the post-typhoon the period.

| Coastal land transformation | Area (km$^2$) | % |
|---|---|---|
| Land replaced by wetland vegetation | 4.06 | 6.67 |

| | | |
|---|---|---|
| Land replaced by wetland | 4.59 | 7.54 |
| Land eroded by water | 7.23 | 11.88 |
| Land accreted | 10.05 | 16.52 |
| Wetland accreted | 2.82 | 4.64 |
| Wetland vegetation replaced by water | 2.12 | 3.48 |
| Wetland replaced by wetland vegetation | 24.17 | 39.71 |
| Wetland replaced by water | 4.41 | 7.25 |
| Water replaced by wetland vegetation | 1.41 | 2.32 |

On the other hand, the short-term effects of a typhoon on the shoreline have also been determined based on the NSM model. The results exhibit the extensive shoreline alteration in the entire Mokpo coastal region after one year of typhoon Soulik, with an accretion of 48.03% transects and erosion of 51.97%. The NSM statistics showed an average shoreline movement of -1.08m, with a recorded mean erosion of -9.25 and deposition of 7.75m (Table 10). The overall erosion was recorded in response to typhoon Soulik even after one year along the Mopko coastal region. This is due to the extensive damage to wetland vegetation during the typhoon period (Table 7). In addition, it was observed that the wetland experience accretion during the typhoon period, but it made the coastline vulnerable to erosion in the near future. The natural native vegetation and wetland vegetation play a critical role in the shoreline stability of the coastal region due to its anti-erosive nature. This phenomenon was evident in the NSM statistics obtained during the post-typhoon period. Therefore, the use of these models can help predict how the shoreline and adjacent coastal landforms will respond to typhoons, identify vulnerable areas, and inform recovery efforts. This can enhance the area's resilience to natural disasters and reduce the risk of future erosion and other environmental problems.

Table 10. Post-typhoon shoreline change statistics based on the NSM model.

| NSM statistics | Summary |
|---|---|
| Total transects | 38313 |
| $NSM_{mean}$ | -1.08m |
| $NSM_{mean\ accretion}$ | 7.75 |
| $NSM_{mean\ erosion}$ | -9.25 |
| $NSM_{maximum\ accretion}$ | 44.76 |
| $NSM_{maximum\ erosion}$ | -121.14 |
| Total transect that records accretion | 18400 |
| Total transect that records erosion | 19913 |
| % of total transect that records accretion | 51.97 |
| % of total transect that records erosion | 48.03 |
| Overall pre to post-typhoon trend | Erosion |

**Comment 4:** The unit of area in Table 7 should be checked.

**Response:** Thank you for the comment. We have reviewed and updated the unit of area in Table 7 in the revised manuscript.

**Comment 5:** The table format is not correct. Check it out in its entirety. Text alignment in table should be checked.

**Response:** Thank you for the comment. We have carefully reviewed the format of all tables in the revised manuscript and made updates wherever necessary.

**Comment 6:** The position of the legend is not correct for each Figure(Figure 4, 8, 9).

Response: As suggested, the position of the legend of Figures 4, 8, and 9 (now Figs. 5, 12, and 13) has been updated in the revised manuscript as,

[Figure]

Figure 5. Status of vegetation greenness based on the NDVI data for the (a) pre-Soulik (01st August 2018) and post-Soulik (15th October 2018) period.

[Figure]

Figure 12. Relative SSC for (a) pre-typhoon and (b) post-typhoon period, while (c) represents the changes in the NDSSI.

[Figure]

Figure 13. The simulated SSC distribution for the surface water of (a) pre-typhoon, (b) post-typhoon period, and (c) represents the spatial changes of SSC from pre- to post-typhoon.

**Comment 7:** The detailed title in Figure 11 should be modified for improvement.

**Response:** Thank you for your comment. The figure caption (Figure 15, revised figure number) has been updated in the revised manuscript as

Figure 15. Net surface area changes (i.e., erosion and accretion) due to typhoon Soulik along the Mokpo coast. Subplots (a-d) show extensive accretion, while erosion is shown in plot (e). The bar graph (f) represents the area changes from the pre to post-typhoon period.

---

## Author Response (AR2)

**Manuscript number: nhess- 2022-253**

My co-authors and I would like to express our gratitude to the reviewers for their constructive feedback and suggestions for strengthening our research. The changes we have made to the attached file in response to such feedback and suggestions have been highlighted in blue to facilitate their identification. I would also like to offer my apologies for the length of time it took us to prepare this response. We also record our deep appreciation for the efficient handling of the manuscript.

**Response to Reviewer#1**

**Overall Observations:** accepted as is.

We are extremely grateful for your valuable feedback and suggestions in your previous comments, which significantly contributed to improving this manuscript. We sincerely appreciate your recommendation to accept the manuscript in its current form without any additional modifications.

**Response to Reviewer#2**

**Overall Observations:** accepted subject to minor revisions.

Thank you very much for your previous comments and suggestions that helped us improve this manuscript. We sincerely appreciate your recommendation to accept this manuscript with minor corrections.

**Comment 1:** Overall comments (figure quality, data source, etc.) were well reflected.

Response: Thank you very much; your remarks are incredibly motivating.

**Comment 2:** Figure (2) However, satellite images at 1-year intervals are not appropriate to consider the effects of short-term erosion caused by typhoons. Therefore, define short, medium, and long-term erosion and specifically state limitations of the study and model.

Response: Thank you for your insightful comment. Short-term erosion refers to the rapid erosion processes and coastal alterations that occur immediately after typhoons or over short durations, typically within days, weeks, or months. Contrarily, medium-term coastal change refers to erosion processes and coastal changes that take place over a period of time ranging from a few months to a few years. It involves the restoration and stabilization of coastal land surfaces after the typhoon. Further, long-term erosion refers to coastal erosion that occurs over extended periods, usually spanning several years to decades or even centuries, influenced by various factors like climate change, land use practices, geological processes, rising sea levels,

and tectonic movements.

The present study addresses the typhoon Soulik-induced morphodynamics over the Mokpo coast region, specifically examining short and medium-term coastal changes. To achieve this, we analyzed pre (2018-08-01) and post (2018-10-15) typhoon Sentinel-2B MSI images to understand the immediate effects of typhoon Soulik, which made landfall near Mokpo City on August 23, 2018. After that, we analyzed the recovery status, i.e., medium-term coastal changes of the Mopko coastal region, using the NSM and coastal landform change model. For this purpose, another Sentinel-2 MSI image was downloaded for the month of October 2019 (one year after the typhoon), as listed in Table 1. Finally, the results exhibited that the proposed model successfully identified the morphodynamic changes caused by the typhoon in the short-term and the recovery status of the coastal landforms in the medium-term.

The above has been discussed in the revised manuscript in Section 3.2, Line No. 217-222. In addition, as suggested, we incorporate the terms 'short-term' and 'medium-term' in the revised manuscript. We also include the limitations of the present study in the conclusion section (Line No. 783-785).